# Dietary Phytase and Lactic Acid-Treated Cereal Grains Differently Affected Calcium and Phosphorus Homeostasis from Intestinal Uptake to Systemic Metabolism in a Pig Model

**DOI:** 10.3390/nu12051542

**Published:** 2020-05-25

**Authors:** Julia C. Vötterl, Jutamat Klinsoda, Qendrim Zebeli, Isabel Hennig-Pauka, Wolfgang Kandler, Barbara U. Metzler-Zebeli

**Affiliations:** 1Institute of Animal Nutrition and Functional Plant Compounds, Department for Farm Animals and Veterinary Public Heath, University of Veterinary Medicine Vienna, Vienna 1210, Austria; Julia.Voetterl@vetmeduni.ac.at (J.C.V.); Jutamat.Klinsoda@vetmeduni.ac.at (J.K.); Qendrim.Zebeli@vetmeduni.ac.at (Q.Z.); 2Institute of Food Research and Product Development, University of Kasetsart, Bangkok 10900, Thailand; 3Field Station of Epidemiology, University of Veterinary Medicine Hannover, Foundation, 49456 Bakum, Germany; Isabel.Hennig-Pauka@tiho-hannover.de; 4Department of Agrobiotechnology, University of Natural Resources and Life Science, Vienna, IFA-Tulln, Tulln 3430, Austria; wolfgang.kandler@boku.ac.at; 5Institute of Physiology and Biophysics, Unit of Nutritional Physiology, Department of Biomedical Science, University of Veterinary Medicine Vienna, Vienna 1210, Austria

**Keywords:** phytase, lactic acid soaking, growing pigs, calcium, phosphorus, gene expression, intestine, kidney, bone

## Abstract

High intestinal availability of dietary phosphorus (P) may impair calcium (Ca) homeostasis and bone integrity. In the present study, we investigated the effect of phytase supplementation in comparison to the soaking of cereal grains in 2.5% lactic acid (LA) on intestinal Ca and P absorption; intestinal, renal, and bone gene expression regarding Ca and P homeostasis; bone parameters; and serum levels of regulatory hormones in growing pigs. Thirty-two pigs were randomly assigned to one of four diets in a 2 × 2 factorial design in four replicate batches for 19 days. The diets comprised either untreated or LA-treated wheat and maize without and with phytase supplementation (500 phytase units/kg). Although both treatments improved the P balance, phytase and LA-treated cereals differently modulated gene expression related to intestinal absorption, and renal and bone metabolism of Ca and P, thereby altering homeostatic regulatory mechanisms as indicated by serum Ca, P, vitamin D, and fibroblast growth factor 23 levels. Moreover, phytase increased the gene expression related to reabsorption of Ca in the kidney, whereas LA-treated cereals decreased the expression of genes for osteoclastogenesis in bones, indicating an unbalanced systemic availability of minerals. In conclusion, high intestinal availability of dietary P may impair Ca homeostasis and bone integrity.

## 1. Introduction

The majority of body phosphorus (P) can be found in bones and teeth, where it forms hydroxyapatite crystals with calcium (Ca) [1,2]. Subsequently, inadequate dietary P intake can negatively impact bone integrity and contribute to diseases like osteoporosis, rickets, or osteomalacia [3,4]. Aside from its function related to the skeleton, phosphates are involved in a plethora of cellular processes, such as enzyme activity, cell signaling, or blood buffering. The most important organs involved in the P homeostasis regulating P absorption, excretion, and storage are the gastrointestinal tract, kidney, and bone [1,2,5]. Phosphorus and Ca homeostasis are closely linked as they are regulated by the same hormones, including vitamin D (VitD), parathyroid hormone (PTH), and fibroblast growth factor 23 (FGF23) [6,7,8,9].

Studies in mice and humans showed that the phosphaturic hormone FGF23 is primarily involved in P homeostasis and is produced by osteoblasts and osteocytes in the bone as well as by pericyte-like cells in the bone marrow as a response to high blood P and mainly suppresses P reabsorption and VitD activation in the kidney [10,11,12]. Moreover, there is evidence from studies in mice, chickens, and humans that FGF23 is expressed in other tissues, including the stomach, small intestines, and lymph nodes [11,13,14]. However, little is known of whether serum levels of the aforementioned regulatory factors correspond to intestinal Ca and P absorption as well as to bone and renal metabolism, especially in young growing adolescents. Moreover, there is evidence from studies in rodents, cows, and pigs that the transcellular and paracellular uptake in the intestine of Ca and P are linked and that the Ca and P availability and their ratio in the diet and digesta influence their reciprocal intestinal uptake [15,16,17]. It was shown that enhanced intestinal availability and absorption of P negatively impact Ca metabolism and if leading to an P overload also negatively impact P metabolism. This is in particular the case when calcium intake is low to moderate as it is the case with phosphate-rich Western-style diets [18,19,20,21].

With regards to dietary sources, cereals and legumes are important sources of P for omnivore and herbivore species as well as for humans. A shift in human nutrition towards a healthier diet replacing meat with cereals and legumes is widely discussed to improve the overall nutritional quality of the diet. However, the majority of P in cereals is bound in the form of phytate-P, which has low digestibility in humans and monogastric animals because the gastrointestinal tract does not express the respective enzymatic capabilities for phytate hydrolysis [22,23,24,25]. By contrast, phytases are expressed by intestinal microbes, which renders phytate-bound P more available to the host, especially in species that rely on alloenzymatic digestion [26,27,28]. In addition, the dietary Ca and P availability modifies the microbiota composition along the digestive tract [29,30]. In order to improve the phytate-P availability and to avert its potential negative impact on the intestinal availability of other minerals like Ca and protein through the formation of complexes with phytate, different food processing techniques have been applied. Traditional food processing methods used in human nutrition, which are also applied in animal nutrition, like soaking, malting, and fermenting of cereal grains, activate endogenous phytase in the germ [23,31,32]. For instance, this technique is applied in bread making using sourdough fermentation or beer brewing or soaking of sorghum [33,34,35]. An adaptation of these methods to make the phytate-degrading effect more controllable is the soaking of grains in mild organic acid solutions, such as lactic acid (LA). In using this method, we recently showed that the phytate-P content of maize, barley, and wheat of 2.2, 2.5, and 3.2 g/kg dry matter pre-incubation, respectively, decreased by 24.4% to 31.4%, which was accompanied by a similar increase in lower *myo*-inositol phosphates. Aside from phytate-P, the soaking of cereal grains in LA also altered the content of other nutrient fractions in the grains, including cations (e.g., Ca), starch, hemicellulose, and protein [36,37]. Whether LA soaking of cereals, however, affects the intestinal, renal, and bone metabolism and/or improves mineral retention has not been sufficiently elucidated yet. Since imbalances in the dietary supply and availability affects the intestinal absorption and body metabolism of Ca and P, we hypothesized that an increased intestinal phytate-P and Ca release due to the LA treatment of cereals may enhance the apparent absorption and retention of Ca and P. We further hypothesized that the changes in absorption and retention of Ca and P should be reflected in increased bone formation, reduced renal excretion of Ca and P, as well as in the serum profile of regulatory hormones.

In order to test our hypothesis, we aimed to create different amounts of intestinally absorbable and subsequently systemically available P, either by soaking cereal grains in a 2.5% LA solution prior to feeding and/or by adding microbial phytase to the diet [23,38,39]. Although phytase is not typically used in human food preparations, by adding phytase to our diets, we could achieve different intestinal and body availabilities of P without changing the total P content of our diets. In fact, the total P content was relatively similar across diets, which minimized the interference from different dietary P intakes on the intestinal and body Ca and P homeostasis. By using a 2 × 2 factorial design, we investigated the effects of phytase supplementation and LA treatment of cereals on the intestinal uptake and retention of Ca and P; intestinal, renal, and bone gene expression related to Ca and P homeostasis; bone parameters; as well as on the serum levels of regulatory hormones in growing pigs. We used the pig as a model as its digestive physiology and mineral metabolism are relatively similar to humans [40,41]. Since more evidence is available for adults [42,43,44], we were especially interested in the role of FGF23 as a regulatory factor for P homeostasis in the growing phase.

## 2. Materials and Methods

### 2.1. Ethical Statement

All procedures involving animal handling and treatment were approved by the institutional ethics committee of the University of Veterinary Medicine Vienna and the National authority according to the Law for Animal Experiments, Tierversuchsgesetz (TVG; BMWFW-68.205/0158-WF/V/3b/2016).

### 2.2. Experimental Design

This study was designed as 2 × 2 factorial arrangement with four experimental diets and four replicate batches. In total, 32 castrated male growing pigs (Large White, age: 6–8 weeks, initial body weight: 13.1 ± 2.3 kg) were randomly allocated to the four diets (*n* = 8/diet). In each replicate, two pigs received one of the four experimental diets. The pigs were housed individually in stainless steel metabolism cages (1.00 m × 1.20 m) with plexiglass walls for visual contact. Pigs had free access to demineralized water. Heating lamps were provided in each cage to keep the ambient temperature between 22 and 24 °C. Each run lasted 19 days with two additional days prior to the start of the experiment for the acclimatization of the pigs to the new environment. Overall, the pigs did not present symptoms of disease or discomfort throughout the experiment.

Phytase (VM Phytase XP 897420, Garant GmbH, Pöchlarn, Austria) was added at the recommended inclusion level of 500 phytase units (FTU)/kg complete diet [39,45,46]. The LA (MILCHSÄURE 80% FOOD KANISTER 25KG, Brenntag Austria GmbH, Wien, Austria) treatment of cereals consisted of soaking whole wheat and maize grains separately in 2.5% LA solution (per batch 8 kg cereal in 12.8 l LA solution) for 48 h at room temperature. After discarding the soaking solution and drying for 24 h (1 h at 70 °C and 23 h at 60 °C), cereals were ground with a cutting mill equipped with a 5-mm sieve (Universal-Schneidemühle Pulverisette 19, Fritsch-GmbH, Idar-Oberstein, Germany). The selected soaking conditions are based on the results of our previous study, showing the greatest decrease in phytate-P [37]. The following diets (Table 1) were formulated: 1) Diet with untreated maize and wheat and no phytase (Con diet); 2) diet with untreated maize and wheat and phytase (Con-Phytase diet); 3) diet with LA-treated maize and wheat and no phytase (LA diet); and 4) diet with LA-treated maize and wheat and dietary phytase (LA-Phytase diet). The content of Ca and P was slightly lower than the nutritional recommendations, whereas the dietary Ca to P ratio was kept at 1.2:1 in the range as recommended by NRC [47], ensuring that only the intestinal availability and not the mineral ratio would influence intestinal and systemic Ca and P metabolism. Every ingredient (wheat, maize, soybean meal, oil, premix, limestone, monocalcium phosphate) was weighed separately prior to each meal and mixed at mealtime. The pigs were fed a freshly prepared mash diet (diet: water = 1:1.15) three times daily at 08.00, 12.00, and 16.00 hours. The daily feed allowances were adjusted to the body weight development and corresponded to 3 times the maintenance requirement ([(body weight^0.6^ × 197)/238.68] × 3) [47,48]. After each meal, feed refusals and spills were collected to calculate the daily feed intake. Twice during each run, samples from each dietary component were collected for nutrient analysis and stored at −20 °C.

### 2.3. Sample Collection

The body weight of pigs was measured on experimental days 0, 8, and 15, and before intestinal sampling. From day 15 to 17, total feces and urine were collected and recorded for the calculation of the P, Ca, and nitrogen (N) balance. Fresh fecal samples were collected from the slatted floor and tray beneath the cage after each feeding and stored at −20 °C. Urine samples were collected in buckets containing 10 mL of sulfuric acid (Sulfuric acid, puriss. p.a., 95.0–97.0%; Merck KGaA, Darmstadt, Germany) to minimize microbial fermentation. The buckets were weighed after feeding and a 50 mL subsample was taken, kept at 4 °C during urine collection, and afterwards, the pooled samples per pig were stored at −20 °C until analyses.

On days 18 and 19 of the experiment, four pigs were sampled 2 h after receiving their morning meal. After sedation (Narketan, 0.1 mL/kg body weight, Ketamine HCl, Vétoquinol GmbH, Ismaning, Germany; and Stresnil, 0.05 mL/kg body weight, Azaperone, Elanco Deutschland GmbH, Bad Homburg v.d Höhe, Germany), blood samples were collected by cardiac puncture. Immediately afterwards, pigs were euthanized by intracardiac injection with T61 (Embutramide, 0.1 mL/kg body weight, Intervet GesmbH, Vienna, Austria) and exsanguinated. Blood samples were centrifuged at 3000 × *g* for 10 min at 4 °C (Eppendorf Centrifuge 5810 R, Eppendorf, Hamburg, Germany) and the serum was stored at −20 °C until analysis. The abdominal cavity was opened at the linea alba and the liver and the whole gastrointestinal tract were removed and visually checked for signs of inflammation. The respective intestinal segments were clamped. For gene expression analysis, the duodenum (20 cm caudal from the pylorus sphincter), mid-jejunum (30 cm), ileum (20 cm cranial from the ileocecal sphincter), whole caecum, and mid-colon (30 cm) were opened at the mesenterium, emptied from digesta, washed in ice-cold phosphate-buffered saline, and dry-blotted with paper towel. The mucosa was scraped off using microscopic glass slides and snap-frozen in liquid nitrogen before storage at −80 °C. For short-chain fatty acids (SCFA) analysis, gastric, jejunal, ileal, caecal, and mid-colonic digesta were collected, homogenized, and placed on ice until storage at −20 °C. Moreover, an aliquot of the homogenized gastric digesta (liquid and solid phases) was collected and stored at −20 °C to determine the soluble P content. Additionally, a 20-cm segment of the jejunum caudal to the segment used for gene expression analysis was transferred into ice-cold transport buffer [49], which was pre-gassed with carbogen (95% O_2_ and 5% CO_2_), to be used in the Ussing chamber for electrophysiological and permeability marker measurements. The right kidney was removed from the cavity and a subsample at the margo lateralis of the cortex was taken, like the liver, washed in ice-cold phosphate-buffered saline, blotted dry, and cut into small pieces, which were then snap-frozen in liquid nitrogen and stored at −80 °C for subsequent RNA isolation. Right metacarpal bones were collected, and their weight and length measured before they were snap-frozen in liquid nitrogen and stored at −80 °C until analyses.

### 2.4. Analytical Methods

#### 2.4.1. Proximate Analysis

Diet samples were ground to pass a 1-mm sieve (Ultra-Zentrifugalmühle ZM 200, Retsch GmbH, Haan, North Rhine-Westphalia, Germany) before being analyzed for dry matter (DM), crude ash, crude protein, P, Ca, neutral-detergent fiber, acid-detergent fiber, total starch, resistant starch, and non-resistant starch according to previously described methods [37,50]. The contents of tP and Ca were measured after microwave digestion ((MLS-ETHOS plus Terminal 320, Leutkirch, Germany). Calcium was determined using atomic absorption spectrometry (Perkin Elmer UV-VIS, model 4100, Perkin Elmer, Shelton, QT, USA), whereas P was measured using the molybdate-vanadate method [51]. The adapted methods of Van Soest [52] with FiberTherm FT12 (C. Gerhardt GmbH & Co. KG, Königswinter, NorthRhine-Westphalia, Germany) were used for the analyses of neutral- and acid-detergent fibers. The contents of total starch, resistant starch, and non-resistant starch were determined using commercial enzymatic assay kits (K-RSTAR, Megazyme International Ireland Ltd., Bray, County Wicklow, Ireland). All analyses were performed in duplicates.

#### 2.4.2. Content of Phosphorus in the Stomach and Balance Study

Soluble P in the homogenized stomach content (liquid and solid fractions) was measured spectrophotometrically (UV-1300; Shimadzu Corp., Kyoto, Japan) at 405 nm with the molybdate-vanadate method [49]. In feces, the DM, P, Ca, and N and in the urine, P and Ca were determined as described above. The nutrient intake, nutrient excretion in feces and urine, as well as apparent absorption and retention were calculated as the mean of the three days of sampling (experimental days 15 to 17) and expressed as grams per day and per kilogram feed intake.

#### 2.4.3. Intestinal Electrophysiology

Intestinal electrophysiological measurements were performed to a modified protocol as previously described [49,53]. Three successional samples prepared from the 20-cm tube piece were evaluated in parallel. Jejunal tissues were stripped of the outer serosal layer before mounting into the Ussing chambers. The mounted segments had an exposed area of 0.91 cm^2^, were gassed with carbogen (95% O_2_ and 5% CO_2_), and kept at 38 °C using circulating water. The time elapsing between the death of the pig and mounting the tissue pieces into the Ussing chambers was between 15 and 30 min. Each Ussing chamber was connected to a pair of dual-channel current and voltage electrodes (Ag–AgCl), being submerged in 3% agar bridges filled with 3 M potassium chloride [49,53]. After equilibration of 20 min under open-circuit conditions, the tissue was short-circuited by clamping the voltage to zero (0 min). The potential difference (mV), short-circuit current (I_SC_, μA/cm^2^), and transepithelial resistance (R_T_, Ω × cm^2^) were continuously recorded using a microprocessor-based voltage-clamp device and software (version 9.10; Mussler, Microclamp, Aachen, Germany). The tissue conductance (G_T_, mS/cm^2^) was calculated as the reciprocal of the R_T_. At 5 min after voltage clamping, fluorescein isothiocyanate-dextran (FITC-dextran, final concentration: 0.5 mmol/L; Sigma-Aldrich, Schnelldorf, Austria) was added to the mucosal side to assess paracellular permeability. Buffer samples from the serosal and mucosal side were taken at 35 and 65 min after marker addition, whereas buffer samples from the mucosal side were collected at 45 and 75 min to measure flux rates. The FITC-dextran concentrations in mucosal and serosal buffers were analyzed and the mucosal-to-serosal flux rates of FITC-dextran were calculated as previously described [49] Disodium hydrogen phosphate were added at 35 min at the mucosal side to measure changes in I_SC_ and R_T_ in order to assess the absorptive capacity of the tissue samples. At the end of the experiment at 115 min, theophylline (inhibitor of the phosphodiesterase, final concentration, 8 mmol; Sigma-Aldrich, Schnelldorf, Austria) was added to both chamber sides to monitor tissue vitality.

#### 2.4.4. Intestinal Short-Chain Fatty Acids and pH in the Digesta

The SCFAs in the digesta of the stomach, jejunum, ileum, caecum, and mid colon were determined using gas chromatography (GC) using a modified protocol [45]. For this, 1.0 g of digesta sample was mixed with 200 µL of 25% phosphoric acid, 300 µL of internal standard (4-methylvaleric acid, Sigma-Aldrich, St. Louis, USA), and with 1000 µL of double-distilled water for gastric, ileal, and caecal digesta samples, 1500 µL for colonic samples, and 500 µL for jejunal samples. Samples were centrifuged (20,000 × *g* for 20 min) and the clear supernatant was used to measure SCFA on the GC-2010 Plus Capillary GC (Shimadzu Corp., Kyoto, Japan) using a 30 m × 0.53 mm × 0.5 µm capillary column (Trace TR Wax, Thermo Fisher Scientific, Waltham, MA, USA) and helium as a carrier gas. The gas chromatograph was equipped with an autosampler and injector (AOC-20s Auto Sampler; AOC-20i Auto-Injector, Shimadzu Corp., Kyoto, Japan) and a flame-ionization detector (BID-2010 Plus, Shimadzu Corp., Kyoto, Japan). Digesta pH was measured with a pH meter (SevenMulti™, Mettler-Toledo GmbH, Columbus, OH, USA).

#### 2.4.5. Serum Parameters

The serum content of total P, total Ca, and alkaline phosphatase (ALP) were determined with enzymatic colorimetric assays using an autoanalyzer for clinical chemistry (Cobas 6000/c501; Roche Diagnostics GmbH, Rotkreuz, Switzerland). Commercial porcine-specific ELISA kits were used to determine FGF23 (Porcine FGF23 ELISA kit; Wuhan Fine Biotech Co., Ltd., Hubei, China). Calcitonin was determined using a porcine-specific ELISA kit (Calcitonin ultrasensitive ELISA; DRG Instruments GmbH, Marburg, Germany); however, most samples contained levels below the quantification limit of 0.7 pg/mL. VitD (25-Hydroxy Vitamin D EIA; Immunodiagnostic Systems Holdings PLC, Tyne & Wear, United Kingdom), OCN (N-MID Osteocalcin ELISA; Immunodiagnostic Systems Holdings PLC, Tyne & Wear, United Kingdom), and PTH (Intact PTH ELISA; Immunodiagnostic Systems Holdings PLC, Tyne & Wear, United Kingdom) were analyzed with commercial ELISA kits for humans that were used in previous pig studies [54,55].

#### 2.4.6. RNA Isolation and Quantitative Real-Time PCR

Total RNA was isolated from 20 mg of frozen intestinal mucosa, kidney, and bone marrow from the cross-section at the middle of the third metacarpal bone using mechanical homogenization and the RNeasy Mini Kit (RNeasy Mini Qiacube Kit (250), Qiagen, Hilden, Germany) as described in Metzler-Zebeli et al. [53]. RNA isolates were treated with DNase I (Turbo DNA-free Kit; Ambion, Austin, TX, USA) to remove genomic DNA. RNA was quantified with the Qubit HS RNA Assay kit on the Qubit 2.0 Fluorometer (Qubit RNA HS Assay Kits/Qubit Fluorometer; Life Technologies Limited, Foster City, CA, USA), and the quality of the extracted RNA was evaluated on the Agilent Bioanalyzer 2100 (Agilent RNA 6000 Nano Assay; Agilent Technologies, Santa Clara, CA, USA). Only RNA with RNA integrity numbers >8 was used in the gene expression experiment. Complementary DNA (cDNA) was synthesized using the High Capacity cDNA RT kit following the manufacturer’s protocol (High-Capacity cDNA Reverse Transcription Kits; Life Technologies Limited, Foster City, CA, USA). Primers used for amplification of the target and housekeeping genes were designed using PrimerBLAST (www.ncbi.nlm.nih.gov/tools/primer-blast/). Intestinal mucosal samples were analyzed for the expression of 16 target genes, the kidney for seven target genes, metacarpal bones for seven target genes, and all samples were analyzed for five housekeeping genes (*HKG*) as an endogenous control (β-actin (*ACTG*), hypoxanthine phosphoribosyl-transferase (*HPRT*), glyceraldehyde 3-phosphate dehydrogenase (*GAPDH*), b-2 microglobulin (*B2M*), and ornithine decarboxylase antizyme 1 (*OAZ1*); Appendix A). Three out of the five HKG (*ACTB*, *GAPDH*, *B2M*) were identified to be the most stably expressed HKGs and were used as endogenous controls after an assessment with NormFinder and BestKeeper [56]. Their geometric mean was used for normalization of the raw gene expression data to determine the ΔCq values. The relative gene expression was calculated using the 2^−ΔΔCt^ method [57]. Basically, the geometric mean of the HKGs was subtracted from the raw gene expression data of the targeted genes to normalize them in order to determine ΔCq values. The sample with the highest expression (lowest ΔCq value) of the respective targeted gene was used to calculate the ΔΔCq values. All primers and their respective amplification efficiencies (E = 10^(−1/slope)^) are provided in Appendix A.

#### 2.4.7. Metacarpal Bone Measurements

The bones were thawed at 4 °C for two days. The third and fourth metacarpal bones were separated and the third metacarpal from the right foreleg was used for the measurement [58]. For this, the remaining organic tissues were carefully removed with a surgical blade. Weight and water displacement of the metacarpal bone were measured (Analytical Balance MS205DU, Mettler-Toledo International Inc., Columbus, Ohio, USA) to calculate the density of the bone (Equation (1)) [59]. The external and internal horizontal, as well as the vertical diameters, were measured with a caliper. These diameters were used for the calculation of the geometrical properties cortical wall thickness (Equation (2)), cortical wall area (Equation (3)), and index to estimate the relative ratio of the cortical wall to the total width of the bone (Equation (4)) [60]. The metacarpal bones were afterwards analyzed for their DM content by oven drying at 103 °C for 24 h. The crude ash content was determined by incineration at 580 °C for 24 h after flaming the bones to remove the remaining exterior organic matter [51,61]:Density of bone (g/cm^3^) = (bone weight (g) × density of water (g/cm^3^)/water displacement (g)(1)
Wall thickness^1^ (mm) = [(B+H) − (b+h)]/4(2)
Cortical wall area^1^ (mm^2^) = (π/4 × [(B × H) − (b × h)](3)
Index^1^ = [((H − h)/H) + ((B − b)/B)](4)
b = vertical internal diameter; B = vertical external diameter; h = horizontal internal diameter; H = horizontal external diameter.

Additionally, the content of calcium oxide (CaO) and phosphorus pentoxide (P_2_O_2_) in the ash of metacarpal bones were analyzed with an inductively coupled plasma mass spectrometry (ICP-MS) using a modified protocol [62]. Briefly, 100-mg samples of bone ash were weighed into 50-mL sample containers of polystyrene. Then, 1.5 mL of ultrapure water and 1.5 mL of nitric acid (69%; TraceSELECT, Sigma-Aldrich, St. Louis, MO, USA) were added. After complete dissolution, the containers were filled up to 50 mL on a balance and shaken rigorously. A total of 500 µL of this solution were mixed with 2.5 mL ultrapure water and 3 mL of internal standard solution containing 40 μg/L Sc, 20 μg/L In, and 20 μg/L Tl in 0.5% HNO_3_ (*v/v*) and aspirated. The nuclides ^31^P and ^44^Ca were measured in medium-resolution mode, Rs = 4000, 10% valley definition, with a double-focusing sector field instrument (Finnigan ELEMENT2, Thermo Electron Corporation, Bremen, Germany) equipped with an autosampler CETAC ASX-520, CETAC Technologies, Omaha, NE, USA). The quantitative analysis of the samples was performed by external calibration.

#### 2.4.8. Statistical Analyses

A power test analysis based on Kononoff and Hanford [63] and on recent data for the gene expression, electrophysiology, and mineral balance of pigs [30,53,64] using SAS (SAS 9.4, SAS Institute Inc., Cary, NC, USA) was performed to identify the minimum number of observations (*n* = 8) per dietary treatment required for the present pig experiment. The power test analysis determined that a statistical power of 90% for a sample size of eight and α = 0.05 could be expected to have sufficient power to reject the null hypothesis (H_0_), if H_0_ was false (*p* = 1 − β). After testing the data for normal distribution using the Shapiro–Wilk test and outlier in SAS, data were subjected to ANOVA using the PROC MIXED of SAS. The model accounted for the fixed effects of phytase, LA, and the interaction phytase × LA. The four runs were considered as a random effect. Degrees of freedom were approximated by the Kenward–Roger method and the pairwise comparisons among least-square means were performed using the Tukey–Kramer test. The individual pig was the experimental unit. Data were expressed as least square means ± standard error of the mean (SEM) and differences at *p* ≤ 0.05 and 0.05 < *p* ≤ 0.10 were defined as significance and trend, respectively. Pearson correlation coefficients were calculated for data from the relative gene expression from kidney and metacarpal bones as well as bone and serum parameters and the Ca/P ratio of the diets using PROC CORR of SAS.

## 3. Results

### 3.1. Diets and Animal Performance

Throughout the trial, the pigs remained clinically healthy and did not show signs of sickness or discomfort. Soaking the cereals in 2.5% LA altered the nutrient composition of the grains (Table 1). Due to the soaking, the LA diets comprised 9.8%, 7.0%, and 10.2% less crude ash, Ca, and total P, respectively, compared to the non-treated cereal diets. Likewise, the content of protein, neutral-detergent fiber, and starch was reduced in the LA and LA-Phy diets compared to the diets with non-treated cereals. Due to this, the actual Ca:P ratios were 1.21:1, 1.26:1, 1.20:1, and 1.25:1 for the Con, LA, Con-Phytase, and LA-Phytase diets, respectively. Across the dietary treatment groups, all pigs consumed their daily feed allowances. Feed intake and growth performance were similar among dietary groups. Nevertheless, the LA treatment of cereals tended (*p* = 0.089) to improve the feed conversion ratio by 6.1% (Appendix A).

### 3.2. P Content in the Stomach and Mineral Balances

The analysis of P in gastric digesta showed that phytase supplementation increased the percentage of soluble P by 31.1% compared to without phytase supplementation. The LA treatment of dietary cereals synergistically enhanced this increase further by 44.7% as indicated by the phytase × LA-treated cereals interaction (Figure 1; *p* = 0.001).

Due to the lower nutrient content in the diets containing LA-treated cereals, P intake was 3.9% less in pigs fed these diets compared to the diets with untreated cereals (*p* < 0.001). Due to these differences in the daily P intake, results are presented as the percentage per dietary intake. The LA-treated cereals caused a 27.2% decrease in total P excretion (*p* = 0.007), which was mainly due to the reduced fecal P excretion (*p* = 0.004) and corresponded to an increase in the apparent absorption of P by 21.8% in pigs fed with LA-treated cereals compared to pigs fed the untreated cereals (Figure 2d; *p* = 0.004). This resulted in a 20.3% increase in P retention due to LA-treated cereals compared to the untreated cereals (Figure 2g; *p* = 0.007). Pigs receiving the diets with the phytase supplementation showed a reduced total P excretion by 37.3% compared to the diets without phytase (*p* < 0.001), which corresponded to an increased apparent absorption of P by 32.0% (Figure 2d; *p* < 0.001). While P excreted in feces was reduced (*p* < 0.001), excretion in urine was increased due to phytase supplementation compared to without the supplementation (Figure 2a, *p* < 0.001). Simultaneously, phytase supplementation increased the retention of P by 27.9% compared to without phytase addition (Figure 2g; *p* < 0.001).

The dietary treatments did not modify the intake of Ca. The LA-treated cereals decreased the total excretion of Ca by 29.9% compared to the untreated cereals (*p* = 0.006). Nevertheless, the LA-treated cereals increased the urinary Ca excretion by 67.9% compared to the untreated cereals, whereby the phytase × LA interaction indicated that this was mainly the case without the phytase addition (Figure 2b; *p* = 0.001). The lowered Ca excretion in feces and urine with the phytase supplementation resulted in a reduction in the total Ca excretion by 41.4% compared to without phytase supplementation (*p* < 0.001). Subsequently, phytase supplementation improved the apparent Ca absorption and retention by 13.2% and 39.6%, respectively, compared to without phytase supplementation (Figure 2e,h; *p* < 0.001).

The wider Ca/P ratio in the LA diets reflected the lower P intake with these diets compared to the diets containing the untreated cereals (*p* < 0.001). When calculating the ratios of Ca/P excreted in feces and urine, the phytase × LA interactions indicated that this ratio increased in feces when both treatments were combined, whereas in urine, the LA-treated cereals as a single treatment largely raised this ratio (Figure 2c; *p* < 0.001). With phytase, in turn, the ratio of excreted Ca:P in urine declined compared to without phytase supplementation, irrespective of whether it was fed as a single treatment or combined with the LA-treated cereals. Despite these differences in the latter ratio, only phytase lowered the totally excreted Ca:P ratio compared to without phytase addition (*p* = 0.032). Both phytase and LA-treated cereals decreased the ratio of absorbed and retained Ca:P (Figure 2f,i; *p* < 0.05). The detailed results for the balance study can be found in Appendix A.

### 3.3. Intestinal Electrophysiology in the Jejunum, Short-Chain Fatty Acid Concentration, and pH along the Intestinal Tract

The jejunal electrophysiological parameters showed a trend for a decrease in the basal I_SC_ due to the LA-treated cereals (Table 2; *p* = 0.066) before the addition of Na_2_HPO_4_. After the addition of Na_2_HPO_4_, ∆I_SC_ tended (*p* = 0.059) to increase by 3 times in the jejunal mucosa of pigs fed the LA-treated cereals compared to pigs fed the untreated cereals. The mucosal to serosal flux of FITC-dextran was 11.1% lower in pigs fed the LA-treated cereals but only with phytase supplementation as indicated by the Phy × LA interaction (*p* = 0.030).

With respect to the SCFA, the LA treatment of cereals increased their concentration by 13.3% in caecal digesta compared to the untreated cereals (Table 3; *p* = 0.001). The pH was 14.1% lower in the stomach of pigs fed LA-treated cereals compared to pigs fed the untreated cereals (*p* = 0.007).

### 3.4. Serum Parameters

Serum parameters showed that the phytase supplementation caused a 1.7% decrease in serum Ca compared to without phytase addition (*p* = 0.041), whereas the LA-treated cereals increased it by 9.1% compared to the untreated cereals (Figure 3a; *p* = 0.014). Phytase supplementation increased serum P by 19.5% compared to without phytase supplementation (Figure 3b; *p* < 0.001). The Ca:P ratio in the serum was increased by LA-treated versus untreated cereals, but the effect was diminished, and the ratio was even further decreased due to phytase by 17.5% compared to without phytase supplementation (Figure 3c; *p* = 0.028). The product of the Ca and P content in serum was 24.8% increased due to LA-treated cereals compared to the untreated cereals (Figure 3d; *p* = 0.041). Additionally, phytase increased VitD levels by 19.6% (Figure 3e; *p* < 0.001), whereas it decreased the serum FGF23 by 10.0% compared to without phytase supplementation (Figure 3f; *p* = 0.034). Moreover, the Phy × LA interaction (*p* = 0.045) indicated that the decrease in serum FGF23 due to the phytase supplementation was stronger when the diet contained the LA-treated cereals compared to the non-treated cereal diet. The phytase supplementation tended to increase the serum level of PTH compared to without phytase supplementation (Figure 3g; *p* = 0.092). Osteocalcin and ALP levels in serum were not affected by treatment (Figure 3h,i).

### 3.5. Gene Expression in the Intestinal Tract

Phytase supplementation affected the relative expression of genes for intestinal mucosal Ca and P uptake, cellular transport, signaling, and basolateral Ca transport more strongly than LA-treated cereals (Table 4). Regarding the apical transport of Ca into the enterocytes, phytase supplementation led to a downregulation of the *TRPV5* expression in the caecum by 49.3% compared to without phytase (*p* = 0.037) and, as a trend (*p* = 0.099), in the colon by 59.4%. Moreover, phytase supplementation downregulated (*p* < 0.001) the jejunal, ileal, and colonic expression of *TRPV6* by on average 66.6% compared to without phytase supplementation. This was accompanied by trends (*p* < 0.10) for phytase effects to lower expression of the Ca-binding protein calbindin 1 (*CALB1*) by 28.8% in the caecum and of the basolateral Ca transport protein plasma membrane Ca^2+^ ATPase (*PMCA1b*) by 14.1% in the colon compared to without phytase supplementation. Additionally, phytase supplementation reduced the ileal *VDR* expression by 38.9% (*p* = 0.006) as well as the duodenal (*p* = 0.018), jejunal (*p* = 0.001), and, as a trend (*p* = 0.089), the ileal expression levels of the VitD-inactivating enzyme *CYP24A1* compared to without phytase supplementation. By contrast, the LA treatment of dietary cereals tended (*p* < 0.10) to upregulate the expression of the phosphate transporters *SCL34A1* in the jejunum by 29.1% and of *SLC34A2* in the ileum by 91.9% compared to the untreated cereals. The LA-treated cereals further led to a 44.4%-decreased jejunal expression of *TRPV6* as well as increased expression of *VDR* by 53.2% in the jejunum and of *PMCA1b* expression by 32.0% in the caecum (*p* < 0.050). The LA-treated cereals raised the expression level of *FGF23* by 1010.5% and 101.7% in the ileum and colon, respectively (*p* < 0.05).

The relative expression of genes for tight junction proteins was equally affected by phytase supplementation and LA-treated cereals but mainly in the hindgut (Appendix A). Phytase supplementation downregulated the relative expression of *CDH1* expression by 41.2% and 6.4% in the ileum and caecum and that of *OCLN* and *ZO1* by 13.7% and 8.8% in the colon (*p* < 0.05), respectively, compared to without the phytase supplementation. By contrast, phytase supplementation tended (*p* < 0.10) to increase the expression of *CLDN4* by 22.4% in the duodenum compared to without phytase. Opposite the phytase supplementation, LA-treated cereals tended (*p* < 0.10) to upregulate the relative expression levels of *CDH1* in the caecum by 5.0%, whereas they tended (*p* < 0.10) to downregulate it in the colon by 8.3% and tended (*p* < 0.10) to decrease those of *CLDN4* by 22.2% in the jejunum and colon compared to the untreated cereals. Moreover, the LA-treated cereals increased (*p* < 0.001) the zonula occludens 1 (*ZO1*) expression by 24.1% in the caecum compared to the untreated cereals, whereas in the colon, both treatments, when applied together, decreased the expression of *ZO1* (*p* = 0.037). Only the phytase supplementation influenced the relative expression of mucin genes. Accordingly, phytase tended (*p* = 0.075) to decrease the expression levels of *MUC2* by 21.9% in the duodenum, and by on average 17.0% in the caecum and colon as well as of *MUC4* by 26.5% in the caecum compared to without phytase supplementation (*p* < 0.05). The phytase × LA interaction (*p* < 0.05) for the ileal *MUC2* expression showed that the LA-treated cereals increased the *MUC2* expression but only without the added phytase.

### 3.6. Relative Gene Expression in the Kidney and Metacarpal Bones

In the kidney (Table 5), phytase supplementation upregulated the expression levels of *TRPV5*, *CALB1*, and *VDR* by 162.0%, 128.0%, and 80.5%, respectively, compared to without phytase supplementation (*p* < 0.001). The LA-treated cereals, in turn, upregulated (*p* < 0.001) the renal expression of *CYP24A1* by 61.9% compared to the untreated cereals. In the bone marrow (Table 5), phytase supplementation downregulated *CYP24A1* and *OCN* by 70.9% (*p* < 0.050). Except for *CYP24A1*, LA-treated cereals downregulated the expression levels of all other investigated genes in the marrow of the metacarpal bones, including *FGF23*, *OCN*, *OPG*, *RANKL*, *CYP24B1* (*p* < 0.050), and, as a trend, *VDR* (*p* < 0.10) by 39.5% to 77.0% compared to the untreated cereals.

### 3.7. Metric Parameters in Metacarpal Bones

Both dietary treatments affected the metric measurements, including the thickness, area, and, index of the metacarpal bones (Table 6). As indicated by the phytase × LA interactions, phytase supplementation, and LA-treated cereals when fed as the single treatments decreased (thickness and index; *p* < 0.05) or tended (*p* = 0.053) to decrease (area) the dimension of the aforementioned parameters. When both treatments were combined, the bone thickness, area, and index were not changed. Moreover, without changing the weight and length of the bones, phytase supplementation increased (*p* < 0.001) the ash content of the bones. In contrast, LA-treated cereals tended (*p* < 0.10) to decrease the DM and ash content of the bones compared to the untreated cereals. Nevertheless, the dietary treatments did not affect the concentration of CaO and P_2_O_2_ in the metacarpal bones.

### 3.8. Pearson’s Correlations between Serum Parameters and Metric Metacarpal Bone Parameters and Genes Expression in Metacarpal Bone and Kidney

Pearson’s correlation coefficients to characterize the relationships of parameters for the Ca and P homeostasis in serum with those in bones and kidneys are presented in Appendix A. Only significant correlations (*p* < 0.05) are presented (|r| > 0.35). Serum Ca positively correlated with P (r = 0.53), FGF23 (r = 0.47), and ALP (r = 0.66) in serum. In contrast, serum Ca negatively correlated with the weight of the metacarpal bone (v = −0.40). Serum P positively correlated with serum ALP (r = 0.48) and the renal VDR expression (r = 0.43) but negatively with P_2_O_2_ content in the bone ash (r = −0.35). The ratio of serum Ca: P positively correlated with serum Ca (r = 0.50) but negatively with serum P (r = −0.45). Likewise, the serum Ca×P index positively correlated with Ca (r = 0.88) and P (r = 0.86) in serum. The Ca/P in serum further positively correlated with FGF23 (r = 0.48) but negatively with the CA content in the bone (r = −0.55) and the expression of TRPV5 (r = −0.73), CALB1 (r = −0.69) and VDR (r = −0.50) in the kidney. In contrast serum Ca×P correlated positively with ALP in serum (r = 0.64) and negatively with the metacarpal weight (r = −0.41). While serum VitD positively correlated with the crude ash content of the metacarpal bone (r = 0.46), it negatively correlated with the relative expression of CYP24A1 (r < −0.39), CYP24B1(r < −0.39), and RANKL (r < −0.37) in the bone. In contrast, serum VitD positively correlated with the expression of TRPV5 (r = 0.36), CALB1 (r = 0.48) and VDR (r = 0.57) in the kidney. Serum FGF23 correlated with serum ALP (r = 0.41) and the Ca:P ratio in the diet (r = 0.37) and negatively with the P_2_O_2_ content in the bone ash (r = −0.35). Opposite to serum VitD, the serum content of FGF23 correlated with the renal expression of TRPV6 (r = 0.43) and PMCA1b (r = 0.43). Serum OCN positively correlated with serum PTH (r = 0.41) and TRPV6 (r = 0.37) and CYP24A1 (r = 0.50) expression in the kidney but negatively with RANKL expression in the bone (r = −0.35). Serum PTH correlated with the metacarpal length (r = 0.40) and DM content of the bone (r = 0.40) but negatively with the P_2_O_2_ content in the bone ash (r = −0.40). Interestingly, serum PTH strongly correlated with FGF23 expression in metacarpal bone (r = 0.80).

Pearson’s correlation coefficients that characterize the relationship between diet, bone parameters, and the relative expression of genes related to P and Ca metabolism in the metacarpal bone and kidney are presented in Appendix A. The Ca:P ratio of the diet negatively correlated with the CaO (r = −0.50) and P_2_O_2_ (r = −0.46) content of the metacarpal bone (r = −0.48) but positively with the CYP24A1 expression in the kidney (r = 0.64). Metacarpal length (r = −0.35) and density (r = −0.40) negatively correlated with the CaO content in the bone ash. Furthermore, the DM content of the bone correlated with its ash content (r = 0.57) and with the bone VDR expression (r = 0.43). Moreover, the ash content of the bone correlated with renal VDR (r = 0.61) and TRPV5 (r = 0.62) expression. Relative CYP24A1 and CYP27B1 expression in the metacarpal bone correlated with the expression levels of all other genes in the metacarpal bone except VDR (r < 0.39 to 0.92). Renal expression levels of CYP24A1 correlated negatively with expression levels of OPG (r = -0.41) and RANKL (r = −0.44) as well as of CYP27B2 (r = −0.39) and FGF23 (r = −0.41) in the metacarpal bone. Moreover, metacarpal OCN expression negatively correlated with TRPV5 (r = −0.39), CALB2 (r = −0.36), and VDR (r = −0.37) expression in the kidney. Renal expression of SLCA34A1 further correlated negatively with the contents of CaO and P_2_O_2_ in the bone (r = −0.37 to -0.40) and TRPV6 with CaO (r = −0.47) and CYP24A1 only with the P_2_O_2_ (r = −0.35) content.

## 4. Discussion

In the present study, we used phytase supplementation and LA treatment of cereals to create different amounts of intestinally absorbable and hence systemically available P to study their effect on parameters related to Ca and P metabolism in the intestine, kidney, and bone and their regulatory hormones in a growing pig model. Both, the phytase supplementation and the LA-treated cereal grains, improved the intestinal availability of P as indicated by the higher P solubility in gastric digesta as well as by increased apparent absorption and retention of P. Non-surprisingly, the effect on P absorption and retention was stronger with the phytase supplementation than with the LA-treated cereals. With the increased intestinal and subsequently systemic availability of P with both dietary treatments, we would have expected to observe an anabolic response of regulatory factors related to Ca, P, and bone metabolism along the intestinal mucosa as well as in the serum, kidney, and bones, especially in the young growing pigs. However, serum Ca, P, and hormones as well as renal and bone tissues showed different responses to the dietary treatments compared to those observed at the intestinal level. Soluble P fraction in gastric digesta may hint at diverging Ca to available P ratios in the gastrointestinal tract caused by the single and combined dietary treatments. This may have led to an unbalanced systemic availability of Ca and P due to the dietary treatments, lowering the effective utilization of Ca and P for osteogenesis and increasing their renal excretion. It is known that high intestinal availability and absorption of dietary phosphates can impair Ca homeostasis [67,68]. This assumption would be supported by the ratios of Ca: P calculated for serum, urine, and retention, which were drastically changed by phytase addition and the LA-treated cereals. Notably, physiological effects in the various organs often differed, depending on whether treatments were applied as a single treatment or combined.

Whilst the phytase supplementation mainly raised the dietary P, Ca, and amino acid (protein) availability according to the absorption data, the LA treatment of cereal grains additionally modified the dietary carbohydrate fractions, including the starch and hemicellulose fractions in the wheat and maize grains [37]. LA treatment-related changes in the nutrient composition can be linked to the leaching of these nutrients (e.g., minerals and starch) into the soaking solution, activation of enzymes in the germ (e.g., hemicellulose), and acid hydrolysis (e.g., phytate-P) [36,37]. These changes likely contributed to the rise in the caecal concentration of total SCFA. Increased large intestinal fermentation and SCFA production has been reported to stimulate intestinal cation (e.g., Ca and Mg) absorption. The SCFA lower the luminal pH, which increases mineral solubility [69], thereby increasing the mineral gradient between the luminal and serosal side, and allowing passive mineral transport to increase [70]. This can be explained by the fact that Ca forms insoluble Ca-phosphate complexes at pH above 7 in the intestinal lumen [71,72]. Although we did not find a direct effect of the LA-treated cereals on caecal pH, the increased caecal SCFA concentrations with the LA-treated cereals may point in the direction that fermentation may have contributed to the overall mineral absorption in pigs fed the respective diets.

Under normal physiological conditions, the majority of the absorbed Ca and P should be used for osteogenesis in a fast-growing animal. With the applied feed allowances, pigs gained 600 g/day during the 19-day experimental period, which corresponded to the normal growth rate of a cross-bred meat-type pig of this age [73]. Therefore, if the intestinally absorbed Ca and P led to a systemic overload, the excessive Ca and P should have been excreted via the kidneys as it has been shown for pigs [74,75,76]. Noteworthy, in other animal species, the main regulation routes can differ, like dogs and cats, in which the Ca and P metabolism relies more strongly on regulatory mechanisms in the intestine and bone rather than in the kidney [77,78].

In general, Ca and P are mostly actively taken up via transporter-mediated transcellular pathways in the small intestine of pigs. Nevertheless, passive absorption via the paracellular route is relevant, especially with the increasing intestinal mineral availability shown in pigs and rats [16,79,80]. In the present study, intestinal expression of Ca and P carrier proteins and signaling likely represented the combined effects of a local adjustment to the actual Ca and P concentrations in the respective intestinal segment as well as regulatory actions to adjust systemic availability of Ca and P due to VitD signaling. From the expression levels, the results supported that the main sites of active carrier protein-mediated transport of Ca and P were the duodenum and jejunum. Despite the improved intestinal P availability, the phytase supplementation did not alter the expression levels of the phosphate transporters *SLC34A1* and *SLC34A2*. This may suggest that if the mRNA was translated into functional protein, the phytase supplementation did not modify transcellular P uptake, which potentially explains the increased serum P with the phytase-containing diet. Oppositely, the LA-treated cereals seemed to enhance the small intestinal P absorption, as indicated by the trends for higher *SLC34A1* and *SLC34A2* expression at the jejunal and ileal mucosa, respectively. This observation was supported by the increase in ΔI_SC_ as a response of the jejunal tissue to the added Na_2_HPO_4_ for pigs fed the LA-treated cereals in the Ussing chamber experiment, showing increased carrier-mediated cotransport of sodium ions and consequently of phosphate and thereby confirming greater jejunal P availability and absorption. Moreover, the LA-treated cereals seemed to increase the net absorption of anions, potentially phosphate, as indicated by the more negative inwardly directed I_SC_ [81] in the Ussing chamber experiment. Albeit in a different intestinal segment, it can be speculated whether an increased paracellular Ca uptake mediated by higher caecal fermentation may have contributed to this enhancement in phosphate transporter expression. In line with that, the lower mucosal expression of the Ca channel *TRPV6* in the jejunum of pigs fed the diets containing the LA-treated cereals may be seen as a feedback control mechanism to reduce intestinal Ca uptake [74,82,83]. The same reason may be true for the lower expression of *TRPV6* in the jejunum and ileum as well as of *TRPV5* (cecum and colon) and *TRPV6* (colon) in the large intestine of pigs receiving the phytase supplementation. Although the expression of the *VDR* was not modified in a similar manner, adjustments in the VitD-regulated intestinal Ca uptake may have been further suggested by the lower mucosal expression of *CYP24A1* along the small intestine with phytase, potentially indicating less hydroxylation and thus degradation of active VitD.

To support our assumption with regards to the paracellular permeability and nutrient uptake, the mucosal-to-serosal flux rate of FITC-dextran indicated a largely increased jejunal permeability in pigs fed the LA diet, allowing more molecules to cross the mucosa. Fittingly, the serum Ca levels showed a similar profile as a response to the diets compared to the FITC-dextran flux rates (r = 0.44; *p* = 0.016), supporting a potential link between paracellular Ca uptake and intestinal permeability. Since pigs were on the diet for 19 days, it is difficult to deduce whether these observations may have been either directly related to the LA diet, microbial action of the diet, or indirectly to systemic adaptations, which then may have led to the greater mucosal permeability. However, a greater paracellular jejunal uptake of Ca with the LA diet might be an explanation for the elevated serum Ca in pigs fed this diet. The gene expression results for tight-junction proteins indicated that tight-junction protein expression was influenced by the LA diet, more in the caecal and colonic mucosa, but inconsistently along the small and large intestinal segments. Except for the trend of a downregulated *CLDN4* expression in the jejunum, these data did not correspond to the measured FITC-dextran fluxes. This underlines that the results at the gene expression level do not necessarily explain the functional observations. Nevertheless, there was a general trend that the phytase supplementation downregulated the colonic expression of *ZO1* and *OCLN*. These findings may correspond to the lower *TRPV5* and *TRPV6* expression in the colon with these diets, again hinting at a coupling of trans- and paracellular Ca transport [16,17,84]. Notably, phytase supplementation reduced mucosal *MUC2* (caecum and colon) and *MUC4* expression (caecum). The question arises here whether this observation was due to changes in the microbial community caused by the varying Ca and P availability in these segments [29,85] or in relation to differences in the luminal Ca concentrations [86,87]. Accordingly, the elasticity of the mucus network formation depends on the intestinally available Ca; therefore, lower Ca availability in the large intestine might have led to negative feedback on *MUC2* and *MUC4* expression [87].

For the serum parameters, it is vital to consider that pigs were not fasted when blood samples were collected but were in the resorptive phase two to three hours postprandial. Therefore, changes seen in serum parameters can be partly related to alterations in nutrient absorption and partly to long-term systemic adaptations to the diets. Overall, serum Ca and P levels emphasized the impact of the dietary Ca and P supply on the Ca and P metabolism as indicated by the Ca:P ratios in the serum, feces, and urine. If the dietary Ca supply is low, the body usually compensates this by the activation of VitD to increase intestinal absorption, renal resorption, and mobilization of Ca from the bones [8,43,88]. Pigs consumed similar amounts of feed per meal, which implies similar VitD intake of all pigs. Therefore, changes in the serum levels may be compensatory actions to adjust the serum Ca levels. Accordingly, pigs fed the phytase-containing diets had higher serum VitD levels, which may have caused the lowered *OCN* expression in the metacarpal bones, indicating reduced osteogenesis with the added phytase to balance serum Ca levels in the young growing animals. Likewise, increased renal expression levels of *VDR*, *TRPV5*, and *CALB1* would support an increased VitD-related resorption of Ca in pigs fed the phytase-containing diets. Calcitriol also stimulates FGF23 secretion from osteocytes [89,90]. However, the expression levels of *FGF23* in the marrow of the metacarpal bones and FGF23 levels in serum showed different dietary responses compared to serum VitD. There was a lowering effect of the LA-treated cereals on the *FGF23* expression in the bone, which corresponded to the lower expression of the VitD receptor and hence indicated decreased VitD signaling in the bone with the LA-treated cereals. However, there were no dietary treatment effects on the renal phosphate transporter expression to adjust renal P excretion [89,91], suggesting a different regulation.

FGF23 plays an essential part in P homeostasis, with the main regulating organs being bone, intestine, and kidney [1,2,5]. Therefore, serum FGF23 may reflect regulatory action in relation to P homeostasis in the aforementioned organs. In the current study, serum FGF23 levels corresponded to the serum Ca levels, Ca/P ratio in serum, and urinary Ca excretion, being the highest in pigs fed the LA diet, and therefore seemed to be regulated by Ca rather than by P, which would also be supported by the present correlations. We would have assumed to find less FGF23 in serum of the pigs fed the LA diet in order to moderate the P excretion via the kidneys to enhance the utilization of Ca and P for osteogenesis. Instead, we found an increased urinary Ca excretion with the LA diet. So far, the relation between dietary Ca absorption and serum Ca and FGF23 levels is poorly understood [90]. Before, serum FGF23 has been positively associated with dietary P availability [92,93], which was not the case in our study. Results from human hemodialysis patients, however, support a strong positive association between serum Ca and FGF23 [94], which needs further investigation in order to complete our current understanding with respect to the underlying physiological signaling.

Due to the close correlation of bone-specific ALP and serum ALP, this biomarker is used in diagnosing osteomalacia in humans [95,96]. This disease is characterized by hypophosphatemia, which leads to a compensatory upregulation of ALP in order to increase free P [2,96]. Our findings demonstrated a positive correlation between serum levels of ALP and FGF23, suggesting that two opposing factors for the regulation of the P homeostasis were equally regulated. In fact, FGF23 has been reported to suppress ALP transcription [97]. Interestingly, the metric bone data showed a different picture compared to the gene expression data, indicating a non-efficient use of Ca and P for the bone mass (i.e., thickness and area) but without affecting the actual mineral concentrations in bones, when dietary treatments were fed as single treatments but not when fed together. More efficient utilization of Ca and P with both dietary treatments would be also supported by the serum FGF23 level and the Ca and P retention, which were the lowest and highest when both treatments were combined, respectively. Despite the effect on the ash content due to both treatments, the CaO and P_2_O_2_ content of the bone ash were little affected by the dietary treatments, which shows that measuring the mineral concentrations in bones is not sufficient, even in young growing animals, to reliably predict diet-related alterations in mineral homeostasis and bone integrity. Therefore, the differences in ash content could have been caused by varying the fat and water associated with nutrition because the bones were not defatted for analysis [98].

Taken together, as we assumed, both dietary treatments were efficient in improving the intestinal P availability and P retention in the present fast-growing pig model. Nevertheless, while phytase supplementation improved the retention of both Ca and P and the LA-treated cereals that of P, this came at the expense of reduced bone integrity, showing imbalances in mineral homeostasis, which was only counteracted when both treatments were combined. This may illustrate the sensitiveness of the mineral homeostasis towards imbalances in the dietary Ca and P uptake, particularly if this condition exists for a certain time (here three weeks). According to the present data and correlations, we may assume an imbalanced intestinal Ca and P absorption as the cause for differences in the activation of regulatory hormones and subsequently of the enzymatic machinery in the kidney and bones. Although VitD, FGF23, and ALP have been used as markers for mineral imbalances in the skeleton, only the weak correlation of FGF23 with serum Ca and the dietary Ca:P ratio may be helpful to link the systemic mineral availability with the dietary intake. Moreover, the results suggested that serum VitD levels are a useful indicator to predict the bone ash content and Ca resorption activity in the kidney, at least at the gene expression level, but this was irrespective from serum Ca or P. Lastly, due to the missing link between dietary P availability, serum FGF23, and P homeostasis, the present results also implicate the need for further research in relation to bone integrity markers for growing individuals. However, although growing pigs resemble adolescent humans in many physiological traits, it should be mentioned here that certain species-specific differences may exist, demanding for a verification of the observed relationships between dietary and serum parameters in humans.

## 5. Conclusions

In conclusion, the present results demonstrated the effectiveness of both dietary treatments, phytase and LA treatment of dietary cereals, to improve the intestinal Ca and P availability and P retention using the growing pig model. Despite this fact, dietary phytase and LA treatment of cereals differently modulated the gene expression related to intestinal absorption and renal and bone metabolism of Ca and P. Especially, the results for urinary Ca excretion and the expression of genes related to mineral homeostasis in the bones and kidney showed an inefficient utilization of the absorbed Ca by the body, which may be related to an improper systemic availability of Ca and P. Accordingly, bone integrity was reduced with phytase and LA-treated cereals when fed as a single treatment, whereas the combination of both treatments counteracted this effect. Moreover, the results indicated the need for further research in order to better understand the relation between dietary mineral levels, serum Ca, and FGF23 and to identify appropriate bone integrity markers for young growing animals.

## Figures and Tables

**Figure 1 nutrients-12-01542-f001:**
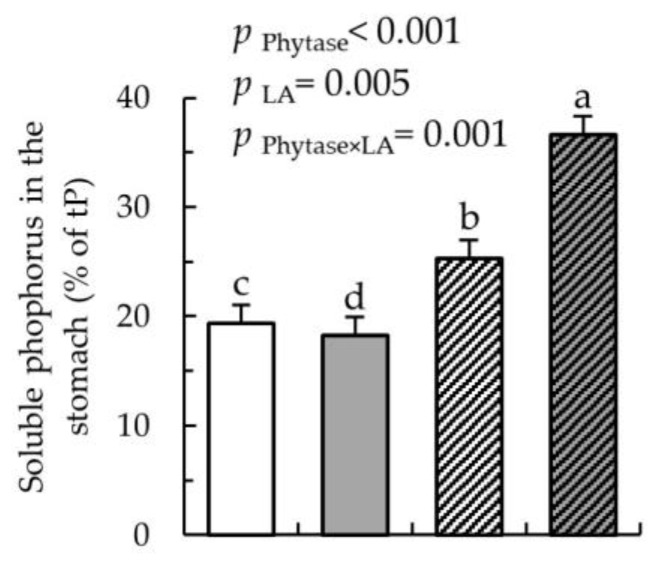
Content of soluble phosphorus (% of tP) in the stomach of pigs fed either Con (
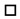
), LA (
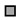
), Con-Phy (
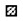
), and LA-Phy (
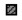
) for 18 days. Values are least square means (*n* = 8/diet), with their SEM represented by vertical bars. Statistically significant (*p* < 0.05) effects of treatment are indicated by different letters (a, b, c).

**Figure 2 nutrients-12-01542-f002:**
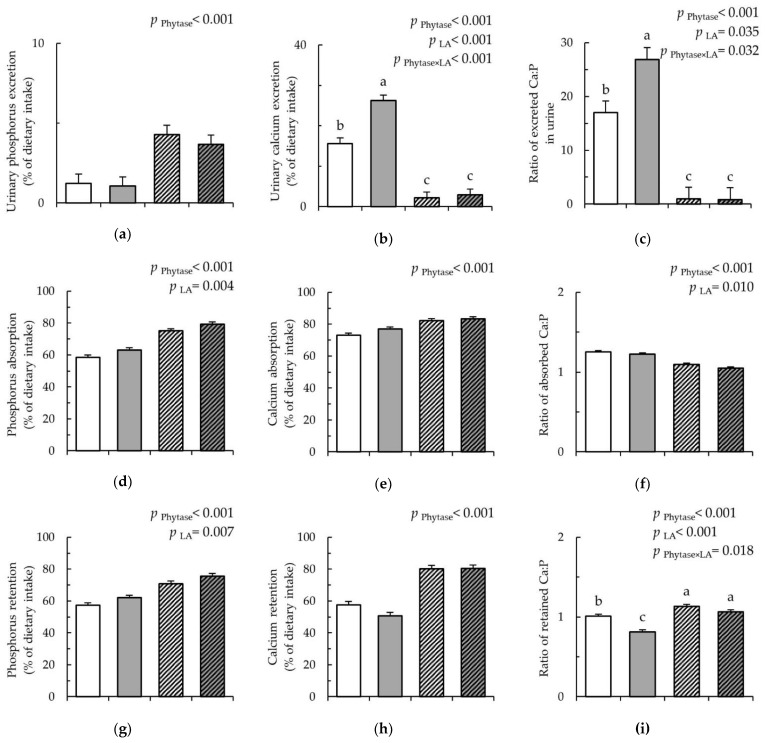
Phosphorus and calcium balance in pigs fed either Con (
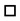
), LA (
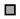
), Con-Phy (
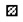
), and LA-Phy (
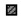
). (**a**) Urinary phosphorus (P) excretion (% of dietary intake); (**b**) urinary calcium (Ca) excretion; (% of dietary intake); (**c**) ratio of excreted Ca:P in urine; (**d**) P absorption (% of dietary intake); (**e**) Ca absorption (% of dietary intake); (**f**) ratio of absorbed Ca:P; (**g**) P retention (% of dietary intake); (**h**) Ca retention (% of dietary intake); and (**i**) ratio of retained Ca:P; Values are least square means (*n* = 8/diet), with their SEM represented by vertical bars. Statistically significant (*p* > 0.05) effects of treatment are indicated by different letters (**a**,**b**,**c**). The nutrient intake, nutrient excretion in feces and urine, as well as absorption and retention were calculated as the mean of the three days of sampling (experimental day 15 to day 17).

**Figure 3 nutrients-12-01542-f003:**
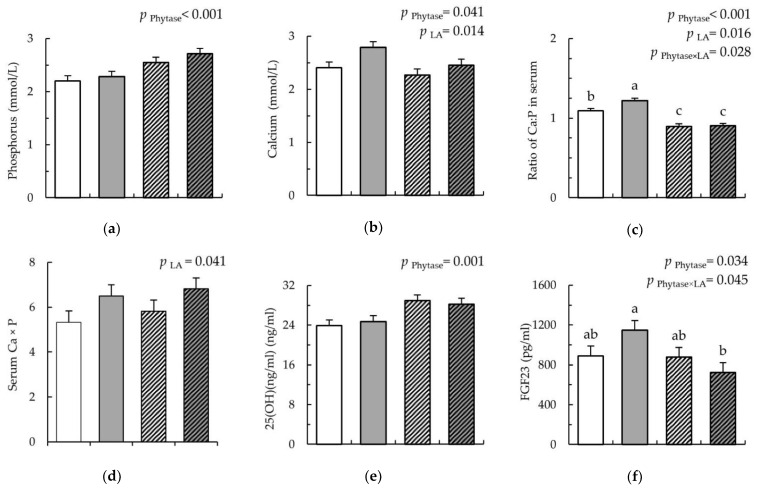
Content of calcium, phosphorus, 25(OH)D3, fibroblast growth factor 23, osteocalcin, alkaline phosphatase, and parathyroid hormone in the serum of pigs fed either Con (
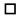
), LA (
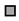
), Con-Phy (
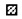
), and LA-Phy (
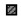
) for 18 days. (**a**) phosphorus (mmol/L); (**b**) calcium (mmol/L); (**c**) ratio of Ca:P in serum; (**d**) product of Ca × P in serum; (**e**) vitamin D (ng/mL); (**f**) fibroblast growth factor (pg/mL); (**g**) parathyroid hormone parathormon (pg/mL); (**h**) osteocalcin (ng/mL); (**i**) alkaline phosphatase (U/L). Values are means (*n* 8/diet), with their standard errors represented by vertical bars. Statistically significant (*p* < 0.05) effects of treatment are indicated by different letters (**a**,**b**,**c**).

**Table 1 nutrients-12-01542-t001:** Diet composition and analyzed nutrient content of the experimental diets.

Dietary Treatment ^1^	Con	LA	Con-Phytase	LA-Phytase
Ingredient (g/kg)				
Wheat	362	-	362	-
Maize	360	-	360	-
LA-treated wheat ^2^	-	362	-	362
LA-treated maize ^2^	-	360	-	360
Soybean meal	220	220	220	220
Oil	20	20	20	20
Premix ^3^ with phytase ^4^	-	-	23	23
Premix ^3^ without phytase	23	23	-	-
Monocalcium phosphate (Ca(H_2_PO_4_)_2_)	4	4	4	4
Limestone (CaCO_3_)	11	11	11	11

Analyzed chemical composition (DM basis, g/kg)				
Dry matter	901	942	900	941
Protein	213	202	216	206
Neutral-detergent fiber	131	122	129	119
Acid-detergent fiber	50	51	52	51
Total starch	519	500	509	493
Non-resistant starch	510	492	500	485
Resistant starch	8.2	7.7	9.2	8.3
Ash	55	50	54	49
Calcium	6.6	6.2	6.6	6.2
Total phosphorus	5.5	4.9	5.5	5.0
Available phosphorus ^5^	1.2	1.6	1.2	1.6
Available phosphorus (%)^5^	21	32	21	32
Phytate-phosphorus ^5^	4.3	3.3	4.3	3.3
Phytate (%) ^5^	79	68	79	68
Metabolizable energy (kJ, kg)	13.7	13.7	13.7	13.7
DCAD ^6^	36	37	36	37

^1^ Con, control; LA, lactic acid-treated cereals. ^2^ LA-treatment implies soaking whole wheat or maize grains in 2.5% LA solution for 48 h. After soaking, solution as discarded and grains were dried and ground before mixing into the diet. ^3^ Provided per kg of complete diet (GARANT GmbH, Pöchlarn, Austria): 15,000 IU of vitamin A; 2000 IU of vitamin D; 14 mg of vitamin E; 2.0 mg of vitamin B1; 6.0 mg of vitamin B2; 3.0 mg of vitamin B6; 0.03 mg of vitamin B12; 4.0 mg of vitamin K3; 30 mg of niacin; 15.0 mg of pantothenic acid; 504 mg of choline chloride; 0.15 mg of biotin; 1.0 mg of folic acid; 40 mg of Mn (as MnO); 90 mg of Zn (as ZnSO4); 90 mg of Fe (as FeSO4); 15.0 mg of Cu (as CuSO4); 0.45 mg of Se (as Na2SeO3), 1.5 mg of iodine (as Ca(IO3)2); 3.52 mg of lysin; 1.20 mg of methionine and cystidin; 1.40 mg of threonine; 0.12 mg of tryptophan. ^4^ 500 FTU/kg diet (VM Phytase XP 897420, Garant GmbH, Pöchlarn, Austria). ^5^ Calculated with data on the average content of available phosphorus and phytate-phosphorus [37,65]. ^6^ DCAD, Dietary Cationen Anionen Difference; calculated with data on the average content of sodium, potassium, chloride, and sulfur [47,66].

**Table 2 nutrients-12-01542-t002:** Effect of dietary phytase and lactic acid-treated cereals on mucosal permeability in the jejunum of pigs mounted in Ussing chambers.

	Dietary Treatment ^1^					*p*-Value		
Parameter ^2^	Con	LA	Con-Phytase	LA-Phytase	SEM	Phytase	LA	Phytase × LA
Na_2_HPO_4_ response								
I_SC_ (µA/cm^2^)	−35.9	−46.3	−31.5	−58.2	9.62	0.705	0.066	0.412
∆ I_SC_	−0.1	0.2	−0.1	0.1	0.13	0.676	0.059	0.729
G_T_ (mS/cm^2^)	8.3	10.8	9.2	9.0	1.06	0.685	0.295	0.204
∆ G_T_	−0.4	−0.6	−0.5	−0.4	0.10	0.718	0.778	0.208
Mucosal to serosal FITC-dextran flux rates								
J_MS_ (mmol/cm^2^ × h)	0.07	0.14	0.07	0.06	0.018	0.069	0.146	0.030

^1^ Experimental period of 19 days; Con, control; LA, lactic acid-treated cereals, Phytase 500 FTU/kg diet (VM Phytase XP 897420). ^2^Values are presented as least square means ± SEM; *n* = 8/diet; I_SC_, short-circuit current; ∆ I_SC_, difference between the maximal I_SC_ value obtained from 2 min after Na_2_HPO_4_ addition and the basal value determined 1 min before Na_2_HPO_4_ addition; G_T_, transepithelial conductance; ∆ G_T_, difference between the basal G_T_ 1 min before Na_2_HPO_4_ addition and the G_T_ value obtained from 2 min after Na_2_HPO_4_ addition; JMS, mucosal to serosal flux rates, FITC-dextran flux rates are calculated by dividing the transport rate (dQ/dt) by the area (A) of the membrane, the transport rate (dQ/dt) was determined by sampling the buffer from the serosal site at 35 and 65 min and from the mucosal site at 45 and 75 min after FITC-dextran addition and analyzing the FITC-dextran content in buffer samples.

**Table 3 nutrients-12-01542-t003:** Effect of dietary phytase and lactic acid-treated cereals on total short-chain fatty acid concentrations (SCFA) and digesta pH along the gastrointestinal tract.

	Dietary Treatment ^1^					*p*-Value		
Paramter ^2^	Con	LA	Con-Phytase	LA-Phytase	SEM	Phytase	LA	Phytase × LA
SCFA (µmol/g)								
Stomach	11.0	11.3	12.8	6.9	2.53	0.608	0.285	0.235
Jejunum	104.6	86.7	87.9	95.4	19.35	0.837	0.789	0.535
Ileum	106.9	51.9	69.4	40.9	26.00	0.379	0.127	0.635
Caecum	169.1	198.6	164.1	184.7	6.88	0.182	0.001	0.529
Colon	172.4	191.1	192.6	187.9	9.36	0.371	0.460	0.223
Digesta pH								
Stomach	4.3	3.9	4.2	3.4	0.20	0.208	0.007	0.258
Jejunum	7.8	7.2	7.4	7.6	0.21	0.898	0.390	0.144
Ileum	8.0	7.9	7.9	7.5	0.30	0.401	0.462	0.676
Caecum	6.3	5.9	6.0	6.1	0.15	0.653	0.428	0.219
Colon	6.6	6.6	6.6	6.6	0.06	0.750	0.392	0.854

^1^ Experimental period of 19 days; Con, control; LA, lactic acid-treated cereals, Phytase 500 FTU/kg diet (VM Phytase XP 897420). ^2^Values are presented as least square means ± SEM; *n* = 8/diet.

**Table 4 nutrients-12-01542-t004:** Effect of dietary phytase and lactic acid-treated cereals on the relative expression of genes (fold change) related to phosphorus and calcium absorption along the intestinal tract.

		Dietary Treatment ^1^					*p*-Value		
Parameter ^2^	Gut Site	Con	LA	Con-Phytase	LA-Phytase	SEM	Phytase	LA	Phytase × LA
*SLC34A1*	Duodenum	0.04	0.03	0.11	0.06	0.037	0.233	0.377	0.627
	Jejunum	0.34	0.41	0.24	0.47	0.077	0.765	0.063	0.330
	Ileum	n.d.	n.d.	n.d.	n.d.	.	.	.	.
	Caecum	n.d.	n.d.	n.d.	n.d.	.	.	.	.
	Colon	n.d.	n.d.	n.d.	n.d.	.	.	.	.
*SLC34A2*	Duodenum	0.73	0.81	0.77	0.59	0.098	0.370	0.628	0.212
	Jejunum	n.d.	n.d.	n.d.	n.d.	.	.	.	.
	Ileum	0.23	0.52	0.14	0.35	0.134	0.363	0.075	0.791
	Caecum	0.06	0.09	0.06	0.07	0.023	0.769	0.455	0.721
	Colon	0.05	0.05	0.03	0.003	0.0018	0.372	0.936	0.936
*TRPV5*	Duodenum	0.18	0.13	0.12	0.12	0.055	0.544	0.669	0.655
	Jejunum	0.29	0.46	0.43	0.39	0.063	0.600	0.305	0.137
	Ileum	0.19	0.23	0.20	0.15	0.045	0.414	0.841	0.305
	Caecum	0.15	0.16	0.06	0.09	0.036	0.037	0.623	0.848
	Colon	0.08	0.06	0.03	0.03	0.022	0.099	0.597	0.779
*TRPV6*	Duodenum	0.23	0.24	0.22	0.25	0.09	0.908	0.512	0.773
	Jejunum	0.60	0.44	0.26	0.22	0.048	<0.001	0.049	0.286
	Ileum	0.025	0.017	0.005	0.004	0.0030	<0.001	0.199	0.224
	Caecum	0.07	0.07	0.04	0.06	0.010	0.227	0.400	0.267
	Colon	0.08	0.08	0.03	0.04	0.009	<0.001	0.690	0.682
*CALB1*	Duodenum	0.43	0.54	0.41	0.39	0.050	0.125	0.385	0.219
	Jejunum	0.23	0.19	0.20	0.13	0.061	0.416	0.374	0.840
	Ileum	0.003	0.005	0.004	0.003	0.0014	0.997	0.703	0.306
	Caecum	0.0001	0.0001	0.0001	0.0001	0.00002	0.075	0.840	0.484
	Colon	0.0006	0.0004	0.0004	0.0005	0.00015	0.801	0.737	0.317
*PMCA1b*	Duodenum	0.04	0.03	0.03	0.03	0.005	0.861	0.332	0.186
	Jejunum	0.22	0.27	0.19	0.32	0.063	0.912	0.165	0.496
	Ileum	0.005	0.006	0.005	0.004	0.0018	0.550	0.989	0.785
	Caecum	0.03	0.04	0.03	0.04	0.002	0.934	0.002	0.119
	Colon	0.04	0.04	0.04	0.03	0.004	0.060	0.260	0.073
*VDR*	Duodenum	0.10	0.08	0.09	0.07	0.010	0.555	0.115	0.739
	Jejunum	0.22	0.26	0.24	0.41	0.054	0.121	0.059	0.217
	Ileum	0.003	0.003	0.002	0.002	0.0005	0.006	0.904	0.347
	Caecum	0.02	0.02	0.02	0.02	0.002	0.146	0.569	0.983
	Colon	0.01	0.01	0.01	0.01	0.001	0.917	0.604	0.588
*CYP24A1*	Duodenum	0.05	0.03	0.01	0.01	0.009	0.018	0.282	0.297
	Jejunum	0.35	0.29	0.06	0.01	0.065	<0.001	0.405	0.989
	Ileum	0.0267	0.0391	0.0004	0.0006	0.01837	0.089	0.734	0.743
	Caecum	0.0004	0.0007	0.0005	0.0009	0.00022	0.614	0.186	0.955
	Colon	0.0011	0.0017	0.0015	0.0016	0.00040	0.711	0.390	0.579
*FGF23*	Duodenum	0.014	0.071	0.001	0.018	0.0335	0.332	0.284	0.550
	Jejunum	0.05	0.10	0.04	0.03	0.034	0.242	0.702	0.386
	Ileum	0.016	0.122	0.002	0.227	0.0691	0.515	0.024	0.398
	Caecum	0.003	0.006	0.002	0.003	0.0028	0.496	0.496	0.708
	Colon	0.008	0.018	0.004	0.013	0.0043	0.301	0.031	0.907

^1^ Experimental period of 19 days; Con, control; LA, lactic acid-treated cereals, Phytase 500 FTU/kg diet (VM Phytase XP 897420). ^2^Values are presented as least square means ± SEM; *n* = 8/diet; *SLC34A1*, Na+-Pi cotransporter 1; *SLC34A2*, Na+-Pi cotransporter 2; *TRPV5*, transient receptor potential vanilloid 5; *TRPV6*, transient receptor potential vanilloid 6; *CALB1*, calbindin; *PMCA1b*, plasma membrane Ca2+ adenosintriphosphatase; *VDR*, vitamin D receptor; *CYP24A1*, cytochrome P450, family 24, subfamily A, polypeptide 1 (24-Hydroxylase); *FGF23*, fibroblast growth factor 23.

**Table 5 nutrients-12-01542-t005:** Effect of dietary phytase and lactic acid-treated cereals on the relative expression of genes (fold change) related to phosphorus and calcium metabolism in the kidney and metacarpal bones.

	Dietary Treatment ^1^					*p*-Value		
Genes of Interest ^2^	Con	LA	Con-Phytase	LA-Phytase	SEM	Phytase	LA	Phytase × LA
Relative gene expression in the kidney								
*SLC34A1*	0.58	0.65	0.63	0.61	0.065	0.998	0.704	0.521
*TRPV5*	0.20	0.15	0.52	0.54	0.043	<0.001	0.636	0.408
*TRPV6*	0.43	0.35	0.30	0.31	0.061	0.184	0.583	0.466
*CALB1*	0.15	0.11	0.31	0.38	0.031	<0.001	0.644	0.085
*PMCA1b*	0.41	0.50	0.47	0.42	0.115	0.912	0.874	0.534
*VDR*	0.39	0.43	0.72	0.70	0.059	<0.001	0.941	0.587
*CYP24A1*	0.38	0.58	0.27	0.65	0.074	0.785	<0.001	0.222
Relative gene expression in the metacarpal bones								
*VDR*	0.23	0.14	0.29	0.09	0.072	0.918	0.058	0.429
*CYP24A1*	0.32	0.21	0.11	0.08	0.064	0.015	0.292	0.547
*CYP27B1*	0.37	0.23	0.29	0.21	0.052	0.361	0.040	0.572
*FGF23*	0.36	0.24	0.52	0.20	0.078	0.436	0.009	0.215
*OCN*	0.43	0.18	0.19	0.06	0.073	0.022	0.016	0.424
*OPG*	0.30	0.09	0.16	0.05	0.063	0.173	0.017	0.440
*RANKL*	0.28	0.15	0.24	0.12	0.054	0.542	0.026	0.899

^1^ Experimental period of 19 days; Con, control; LA, lactic acid-treated cereals, Phytase 500 FTU/kg diet (VM Phytase XP 897420). ^2^Values are presented as least square means ± SEM; *n* = 8/diet; *SLC34A1*, Na^+^-P_i_ cotransporter; *TRPV5*, transient receptor potential vanilloid-5; *TRPV6*, transient receptor potential vanilloid-6; *CALB1*, calbindin; *PMCA1b*, plasma membrane Ca^2+^ adenosintriphosphatase; *VDR*, vitamin D receptor; *CYP24A1*; cytochrom P450, family 24, subfamily A, polypeptide 1 (24-hydroxylase); *CYP27B1*, cytochrome P450, family 24, subfamily B, polypeptide 1 (1α-hydroxylase); *FGF23*, fibroblast growth factor 23; *OCN*, osteocalcin; *OPG*, osteoprotegrin; *RANKL*, receptor activator of NF-κB ligand.

**Table 6 nutrients-12-01542-t006:** Effect of the dietary phytase and lactic acid-treated cereals on the metric parameters of metacarpal bones.

	Dietary Treatment ^1^					*p*-Value		
Paramter ^2^	Con	LA	Con-Phytase	LA-Phytase	SEM	Phytase	LA	Phytase × LA
Length (mm)	5.54	5.54	5.60	5.50	0.166	0.944	0.769	0.769
Weight (g)	27.25	21.38	22.36	22.89	2.777	0.549	0.338	0.260
b (mm)	7.90	8.37	8.35	8.44	0.266	0.341	0.307	0.483
B (mm)	10.84	11.08	10.66	11.32	0.470	0.943	0.346	0.656
h (mm)	8.48	8.70	9.05	8.54	0.324	0.534	0.654	0.266
H (mm)	11.63	11.30	11.58	11.85	0.406	0.539	0.947	0.469
Thickness (mm)	1.52	1.33	1.21	1.55	0.103	0.674	0.482	0.016
Area (mm2)	46.22	41.25	37.77	48.90	3.960	0.921	0.443	0.053
Index	0.54	0.47	0.43	0.53	0.028	0.331	0.656	0.007
Density (g/cm3)	1.16	1.14	1.16	1.16	0.019	0.499	0.584	0.629
DM (%)	55.44	53.76	56.66	54.56	0.920	0.283	0.051	0.822
Ash (%)	16.06	15.28	18.15	17.29	0.420	<0.001	0.063	0.913
Calcium oxide (% of ash)	50.49	48.92	49.85	50.18	0.713	0.671	0.394	0.198
Phosphorus pentoxide (% of ash)	38.55	38.50	38.74	39.26	0.492	0.344	0.634	0.564

^1^ Experimental period of 19 days; Con, control; LA, lactic acid-treated cereals, Phytase 500 FTU/kg diet (VM Phytase XP 897420). ^2^Values are presented as least square means ± SEM; *n* = 8/diet; b, vertical internal diameter; B, vertical external diameter; h, horizontal internal diameter; H, horizontal external diameter; DM, dry matter.

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
