# Peer review of "Dietary Phytase and Lactic Acid-Treated Cereal Grains Differently Affected Calcium and Phosphorus Homeostasis from Intestinal Uptake to Systemic Metabolism in a Pig Model"

_nutrients, 2020, doi:10.3390/nu12051542_

Round 1
Reviewer 1 Report
Dear,
The author found phytase supplementation may impair calcium and phosphate homeostasis in pigs, which leads to bone and kidney diseases.
This study found detrimental effects of dietary phytate hydrolysis on bone and kidney health by supplementation of acid phytase because dietary phytate hydrolysis is beneficial effects of the animal health on bone and kidney.
Consistent with our previous report, this study showed that dietary phytate hydrolysis by phytase might cause renal phosphate wasting and intestinal phosphate overloading, which causes dysregulation of calcium and phosphate homeostasis that leads to bone and kidney disease.
You need to refer the following references in their manuscript.
1) The effects of phytase supplementation are consistent with a recent study for the impact of high phytate diets on the calcium and phosphate homeostasis (eLife 2020;9:e52709). Interestingly, phytase supplementation increased serum levels of vitamin D (the author needs to clarify which vitamin metabolites (25-vitamin D or 1,25-Vitamin D) and decreased serum levels of FGF23, a major regulator of phosphate homeostasis. This finding is consistent with a large population-based study of subjects with preserved renal function without chronic kidney disease, which showed that PTH levels increase earlier than FGF23 levels as kidney function decreases (Kidney International 2016, 90:648–657). Moreover, diets based on vegetable protein lead to lower plasma levels of phosphate and FGF23 compared with those based on meat protein (Clinical Journal of the American Society of Nephrology, 2011 6:257–264). High-phytate with low calcium diet also showed the lower levels of serum FGF23 compared to control (eLife 2020;9:e52709).
2) I wanted the author to address the following comments.
Table 1. they summarized the diet composition and analyzed the nutrient content of experimental diets. Please add the content of dietary phytate percent in the table after simple calculation. It would be beneficial to understand how much phytate present in the diets.
Figures. In each leged, please add the information for the diet period (such as 19 days in the figure legends), although the author pointed out the diet period in the method section. It would be easier to understand for the reader.
Table 4 and 5. It is very hard to understand the fold changes of each gene. Please modify the relative expression fold compared to control (1 vs fold changes in each data), it would be much easier to identify the corresponding expression pattern in each group.
Author Response
The author found phytase supplementation may impair calcium and phosphate homeostasis in pigs, which leads to bone and kidney diseases.
This study found detrimental effects of dietary phytate hydrolysis on bone and kidney health by supplementation of acid phytase because dietary phytate hydrolysis is beneficial effects of the animal health on bone and kidney.
Consistent with our previous report, this study showed that dietary phytate hydrolysis by phytase might cause renal phosphate wasting and intestinal phosphate overloading, which causes dysregulation of calcium and phosphate homeostasis that leads to bone and kidney disease.
You need to refer the following references in their manuscript.
1) The effects of phytase supplementation are consistent with a recent study for the impact of high phytate diets on the calcium and phosphate homeostasis (eLife 2020;9:e52709). Interestingly, phytase supplementation increased serum levels of vitamin D (the author needs to clarify which vitamin metabolites (25-vitamin D or 1,25-Vitamin D) and decreased serum levels of FGF23, a major regulator of phosphate homeostasis. This finding is consistent with a large population-based study of subjects with preserved renal function without chronic kidney disease, which showed that PTH levels increase earlier than FGF23 levels as kidney function decreases (Kidney International 2016, 90:648–657). Moreover, diets based on vegetable protein lead to lower plasma levels of phosphate and FGF23 compared with those based on meat protein (Clinical Journal of the American Society of Nephrology, 2011 6:257–264). High-phytate with low calcium diet also showed the lower levels of serum FGF23 compared to control (eLife 2020;9:e52709).
Authors: First of all, thank you for the helpful comments. As suggested, we included the recommended articles. Reference numbers are 19,20 and 21.
2) I wanted the author to address the following comments.
Table 1. they summarized the diet composition and analyzed the nutrient content of experimental diets. Please add the content of dietary phytate percent in the table after simple calculation. It would be beneficial to understand how much phytate present in the diets.
Authors: Thank you for this suggestion. We added the phytate content of the diets expressed as g/kg and percentage in addition to the available P content in Table 1.
Figures. In each legend, please add the information for the diet period (such as 19 days in the figure legends), although the author pointed out the diet period in the method section. It would be easier to understand for the reader.
Authors: Done accordingly.
Table 4 and 5. It is very hard to understand the fold changes of each gene. Please modify the relative expression fold compared to control (1 vs fold changes in each data), it would be much easier to identify the corresponding expression pattern in each group.
Authors: Thank you for this comment. Of course, it is easier to identify the changes compared to the control when the control is set to 1. However, we used a 2 x 2 factorial design in our study, which means we compared two factors and therefore did not have this “classical” control. This is the reason why we decided to apply the 2^-ddCT method and not the –dCT method to calculate the fold changes.
Moreover, there is another advantage by using the 2^-ddCt method as such that the various gut sites could be compared with respect to their expression level of the respective gene.
Reviewer 2 Report
Dear authors,
thank you for this interesting paper on Ca and P homeostasis in growing pigs.
Please find below my comments on the manuscript.
abstract
- 29 what is meant by “systemic Ca and P levels”? Please find another word for systemic
introduction
- 52 it is known that, for example, P absorption increases PTH, please refer to adequate literature
l.55 "...that Ca and P availability influence intestinal uptake..."
l.52-58 please add the species in which these mechanisms/effects have been shown. There is evidence that there are species differences in Ca and P handling and the importance of different regulatory pathways of Ca and P homeostasis (see e.g. Böswald et al. 2018). So the species needs to be clear for each statement on Ca and P homeostatic regulation.
l.57 be more specific regarding the negative impact of high P availability. It may impact not only Ca, but also P metabolism
l.59 please also add the species (group) - cereals and legumes are important P sources, but especially for omnivores, I´d specify here.
l.63 "..., P more available to the host, especially in species which rely more on alloenzymatic digestion."
l.66 food or feed?
l.70 “decreased phytate-P content of 24.4 to 31.4%...” decreased from what level?
l.74 rephrase “…of the other element” – of the respective other element?
l.76 how will a release of phytate-P increase Ca absorption and retention? Why would increased aD(P) increase aD(Ca)? In the extreme situation, more “free” P in the GIT lumen might even decrease Ca availability by possible Ca-P-complex formation.
l.79ff rephrase sentence or split into 2 sentences for better understanding.
l.86ff After giving the hypothesis and the aim to test it in your study, you then state that you used pigs as model animal for human mineral metabolism. This seems a bit far-fetched because the cereal treatments are used in/intended for farm animal feed, not human food. The general aspect of Ca and P availability and mineral metabolism may be of interest for human physiology/medicine, but in my view you try to put a translational aspect on top of a farm animal nutrition study. To make the comparison pig – human in terms of mineral homeostasis more plausible, this needs to be elaborated further in the introduction with corresponding references.
Also, how should low P supply or low P availability be a problem in human nutrition? In regular human diets, P excess or P supply from certain inorganic additives is more of a problem than low or marginal P supply (see lots of studies by Calvo & Uribarri, for example).
Material & methods
l.255ff third metacarpal from right forelimb was chosen for bone mineral analysis – you might add references for choosing this bone, e.g. Miesorski et al. (2018) Proceedings of the ESVCN congress.
please give the Ca and P supply in relation to recommendations/requirements
l.164 please describe Ca and P analytical methods (not explained in reference no 41)
l.172 was the liquid or solid phase of stomach content sampled or was stomach content homogenized before analysis? This makes a difference for interpretation.
l.172ff why was a different method used for P analysis in stomach content and faeces?
l.175 “apparent absorption”?
- 195 please describe the method for disodium hydrogen phosphate analysis
l.216 total Ca or ionized Ca in serum?
l.219ff were the kits for FGF23, Vit D, PTH species-specific or at least validated for pigs?
Results
Table 1 and rest of manuscript: please be consistent in spelling (analysed vs. analyzed)
Table 1 Are these values means of all batches used? Please give the chemical equation for monocalcium phosphate, too. Please add the energy content, the DCAB and Ca and P content in relation to recommendation/requirement of all 4 diets. Available P should also be expressed as % of total P
l.331f “27.2% decrease in total P excretion…” decrease compared to control? Please clarify, also for the following statements with increased/decreased absorption, excretion and retentions (l. 334ff).
Figure 1: soluble P in % of what – dry matter? total P? Are Con and LA really significantly different? Why do you state significant differences as “p > 0.05”? If it is a trend, then no different superscript letters should be used.
l.333 “apparent absorption”
l.339 retention instead of intake of Ca?!
Figure 2: it would be interesting to correlate faecal Ca excretion and faecal P excretion. There may be a pattern distinguishing the treatment groups, in addition to the Ca/P ratios of absorbed and retained amounts.
Figure 2b) the higher Ca excretion may be due to acidification of the ration with LA addition – was the DCAB of the diets checked? This effect may be worth discussing.
- 293 and e.g. l. 412 you consider differences with a statistic p value of up to 0.1 a trend. A lot of the results in the gene expression section are dominated by trends and not significant differences between the groups. I recommend distinguishing more clearly between the significant results and the other parameters that did not reach significance but showed a trend.
l.382ff suggestion for discussion: if more P is available due to processing, this may stimulate microbial growth and thus increase the SCFA content due to more fermentative activity.
l.392 the serum Ca/P ratio may not be the best parameter to evaluate mineral homeostasis. The serum Ca x P product is a standard with diagnostic value, so it might be worth calculating and comparing this product.
Figure3 titles of y axis: 25(OH)D3 instead of vitamin D, PTH or parathyroid hormone not “Parathormon”
Is Table 4 relevant, if no significant differences are shown? This might be put in supplemental material
Discussion
- 518 The effect of LA on P absorption was only very slight – please rephrase the sentence to make clear that the phytase effect was much higher than the LA affect. Please also strictly refer to apparent digestibility in this case
- 538 how was modification of the CH fractions measured?
l.552 pig model for what? It is not really clear what kind of model you try to generate with the pigs. The feed treatments are not relevant for human nutrition or physiology.
l.554 “excessively” instead of “overly”
l.613ff can the Ca/P ratios in these substrates really indicate homeostasis imbalances?
l.663 + 667 there were no “dramatic imbalances” in Ca and P homeostasis observed
l.670 vit D, FGF23, ALP as “skeletal disease markers” is not a suitable description, please rephrase
l.671 “eating habits…adolescent pig model” – and again, model for what?
Author Response
AUTHORS: Dear reviewer, thank you for the critical but helpful comments to improve our manuscript.
abstract
l.29 what is meant by “systemic Ca and P levels”? Please find another word for systemic
Authors: We modified the sentence for clarity (New Lines 27-30).
introduction
l.52 it is known that, for example, P absorption increases PTH, please refer to adequate literature
Authors: A reference about the main regulators of Ca and P metabolism, vitamin D and PTH next to FGF23 was added accordingly (New Line 47).
l.55 "...that Ca and P availability influence intestinal uptake..."
Authors: This statement was modified for clarity (New Line 58-61).
l.52-58 please add the species in which these mechanisms/effects have been shown. There is evidence that there are species differences in Ca and P handling and the importance of different regulatory pathways of Ca and P homeostasis (see e.g. Böswald et al. 2018). So the species needs to be clear for each statement on Ca and P homeostatic regulation.
Authors: Thank you for this comment. We added the species for the studies describing mechanistic effects throughout the Introduction section (New Line 48-58).
l.57 be more specific regarding the negative impact of high P availability. It may impact not only Ca, but also P metabolism
Authors: Thank you for your comment. The negative impact of high P availability on Ca and P metabolism was added (New Lines 58-61).
l.59 please also add the species (group) - cereals and legumes are important P sources, but especially for omnivores, I´d specify here.
Authors: This sentence was modified accordingly (New Lines 62-65).
l.63 "..., P more available to the host, especially in species which rely more on alloenzymatic digestion."
Authors: Thank you for improving the understanding of the statement (New Lines 68-69).
l.66 food or feed?
Authors: The used processing methods are similar to those applied in human nutrition in order to enhance nutrient values of food (New Lines 73-76).
l.70 “decreased phytate-P content of 24.4 to 31.4%...” decreased from what level?
Authors: We added the initial amount of phytate-P in the grains (New Lines 76-83).
l.74 rephrase “…of the other element” – of the respective other element?
Authors: The wording of the sentence was altered for better understanding (New Lines 85-86).
l.76 how will a release of phytate-P increase Ca absorption and retention? Why would increased aD(P) increase aD(Ca)? In the extreme situation, more “free” P in the GIT lumen might even decrease Ca availability by possible Ca-P-complex formation.
Authors: Thank you for this comment. We agree that a higher amount of “free” P may decrease the intestinal availability of Ca by forming insoluble Ca-P complexes. However, often phytate hydrolysis also results in a simultaneous release of Ca ions, thereby improving their availabilty. We reworded the respective part of the Introduction (New Lines 72, 81-83).
l.79ff rephrase sentence or split into 2 sentences for better understanding.
Authors: This sentence was restructured (New lines 91-94).
l.86ff After giving the hypothesis and the aim to test it in your study, you then state that you used pigs as model animal for human mineral metabolism. This seems a bit far-fetched because the cereal treatments are used in/intended for farm animal feed, not human food. The general aspect of Ca and P availability and mineral metabolism may be of interest for human physiology/medicine, but in my view you try to put a translational aspect on top of a farm animal nutrition study. To make the comparison pig – human in terms of mineral homeostasis more plausible, this needs to be elaborated further in the introduction with corresponding references.
Also, how should low P supply or low P availability be a problem in human nutrition? In regular human diets, P excess or P supply from certain inorganic additives is more of a problem than low or marginal P supply (see lots of studies by Calvo & Uribarri, for example).
Authors: Thank you for this comment. The pig has been used as model also in relation to bone health previously and the suitability of this animal model has been shown (Lüthje et al. 2018 DOI: 10.1177/0023677218766391, Macfayden et al. 2019 10.1186/s12891-019-2452-0, Sobol et al. 2018 DOI: 10.1017/S0007114518000764). It is clear that no animal model (including rodents and sheep) can represent the situation in humans to 100%.Their diets also look different to those of humans and their intestinal metabolism is different (sheep as a ruminating animal and rodents with coprophagia). In any way, in order to improve our understanding in relation to homeostatic regulation of bones, we rely on animal models as we cannot collect the respective samples from humans.
Overall, there must have been a misunderstanding. This study was not designed as a farm animal nutrition study per se. Of course, the diet will still look like a pig diet because certain nutrient requirements need to be met. Nevertheless, cereals and soybean/legume products are a major part of human diets (especially in some regions of the world and communities) and nowadays also particularly in the steadily growing vegan community. There are human legume products that are based on protein- and fiber-rich meals and cakes from the vegetable oil industry (think lineseed meal, pumpkin seed meal, finely ground almond flour and even soybean meal used in baking (https://www.sciencedirect.com/topics/biochemistry-genetics-and-molecular-biology/soybean-meal)). In having said that, similar problems with respect to the available P supply occur in humans and pigs. Due to the addition of phosphates to processed human foods, there is this problem with the too high intake of phosphates. But at the same time there is this problem with a high phytate-P intake in people that consume a plant-based (raw) diet with very low or no animal product intake. Hence, both problems – too low and too high available P intake - occur in humans.
We added several examples from human food preparations and explained the reason for using phytase in the Introduction section. We used phytase as a tool to modulate the intestinal P availability without changing the total P content of the diet. Changes in the total P content of the diet would have biased the outcome of our study by representing an additional factor.
Material & methods
l.255ff third metacarpal from right forelimb was chosen for bone mineral analysis – you might add references for choosing this bone, e.g. Miesorski et al. (2018) Proceedings of the ESVCN congress.
please give the Ca and P supply in relation to recommendations/requirements
Authors: Thank you for mentioning this research work to better justify the regularly used metacarpal bone as representative sample for the skeletal system. The information about Ca and P supply in the diet was added in the description of the Experimental design. (New Lines 277, 131-133)
l.164 please describe Ca and P analytical methods (not explained in reference no 41)
Authors: This was modified accordingly (New Lines 182-186).
l.172 was the liquid or solid phase of stomach content sampled or was stomach content homogenized before analysis? This makes a difference for interpretation.
Authors: This information was added (New Lines 167-168 and 193).
l.172ff why was a different method used for P analysis in stomach content and faeces?
Authors: Apologies for the confusing description. The same methods were used to determine P in the various samples (feed, faeces and stomach content) (New Lines 182-186).
l.175 “apparent absorption”?
Authors: “Apparent” was added to specify the description (New Line 197).
195 please describe the method for disodium hydrogen phosphate analysis
Authors: Apologies, there was a mistake in the description of the method, which has been corrected (New Lines 217-219).
l.216 total Ca or ionized Ca in serum?
Authors: Total Ca content was analyzed in the serum. This information was added (New Line 237).
l.219ff were the kits for FGF23, Vit D, PTH species-specific or at least validated for pigs?
Authors: The missing information about the ELISA kits were added (New Lines 241-247).
Results
Table 1 and rest of manuscript: please be consistent in spelling (analysed vs. analyzed)
Authors: Thank you for noticing the inconsistency of spelling, this was corrected.
Table 1 Are these values means of all batches used? Please give the chemical equation for monocalcium phosphate, too. Please add the energy content, the DCAB and Ca and P content in relation to recommendation/requirement of all 4 diets. Available P should also be expressed as % of total P
Authors: Chemical equations were added. The information about Ca and P content in relation to recommendations was added in the diet description (New Lines 131-133). The information about the amount of available P and phytate-P in g/kg as well as in % and DCAD values was added to Table 1.
l.331f “27.2% decrease in total P excretion…” decrease compared to control? Please clarify, also for the following statements with increased/decreased absorption, excretion and retentions (l. 334ff).
Authors: In this study we used a 2x2 factorial design. Therefore, there is not this “classical” control treatment. Nevertheless, we modified the Results section by indicating which factor combinations (with versus without phytase and LA-treated versus untreated cereals) were compared. (New lines 346-386).
Figure 1: soluble P in % of what – dry matter? total P? Are Con and LA really significantly different? Why do you state significant differences as “p > 0.05”? If it is a trend, then no different superscript letters should be used.
Authors: Thank you. The missing information was added (Figure 1).
L.333 “apparent absorption”
Authors: The terminology was adapted (New lines 346-386).
l.339 retention instead of intake of Ca?!
Authors: There must have been a misunderstanding. The intake of Ca was correct in this sentence. We split this paragraph into 2 to make it clearer that P and Ca balance were separately described (New Lines 368-376 for the Ca balance).
Figure 2: it would be interesting to correlate faecal Ca excretion and faecal P excretion. There may be a pattern distinguishing the treatment groups, in addition to the Ca/P ratios of absorbed and retained amounts.
Figure 2b) the higher Ca excretion may be due to acidification of the ration with LA addition – was the DCAB of the diets checked? This effect may be worth discussing.
Authors: Thank you for this suggestion. The Ca/P ratios for all parameters (excretion, urinary excretion, faecal excretion, absorption and retention) can be found in Table S3. The correlation of balance parameters did not improve the understanding of these parameters. The DCAD values were calculated, but the difference among diets is negligible and was therefore not discussed.
l.293 and e.g. l. 412 you consider differences with a statistic p value of up to 0.1 a trend. A lot of the results in the gene expression section are dominated by trends and not significant differences between the groups. I recommend distinguishing more clearly between the significant results and the other parameters that did not reach significance but showed a trend.
Authors: The differentiation between trends and significances in the text was improved (throughout Results section).
l.382ff suggestion for discussion: if more P is available due to processing, this may stimulate microbial growth and thus increase the SCFA content due to more fermentative activity.
Authors: Thank you for this suggestion. Indeed, with more P available, this may have stimulated the fermentation. However, the effect on cecal SCFA concentrations was similar for the LA diet and the LA-phytase diet, showing that there was a general LA-treatment of cereal-effect. There was no additional effect of the extra P originating from phytase release as indicated by the missing phytase x LA-treated cereal interaction. Therefore, the effect on cecal SCFA was probably due to changes in other nutrient fractions caused by the LA-treatment of cereals, such as the starch and hemicellulose fractions. This was discussed accordingly and can be found in the Discussion section in Lines 589-603.
l.392 the serum Ca/P ratio may not be the best parameter to evaluate mineral homeostasis. The serum Ca x P product is a standard with diagnostic value, so it might be worth calculating and comparing this product.
Authors: The serum Ca x P was calculated and Ca/P as well as Ca x P was included in the correlation and Ca/P in serum showed a correlation with FGF23 whereas Ca x P showed a correlation with ALP (Table S5).
Figure3 titles of y axis: 25(OH)D3 instead of vitamin D, PTH or parathyroid hormone not “Parathormon”
Authors: Corrected (Figure 3).
Is Table 4 relevant, if no significant differences are shown? This might be put in supplemental material
Authors: There must have been a misunderstanding. There are significant effects in relation to Ca and P transport processes along the intestinal tract (Table 4).
Discussion
518 The effect of LA on P absorption was only very slight – please rephrase the sentence to make clear that the phytase effect was much higher than the LA affect. Please also strictly refer to apparent digestibility in this case
Authors: The sentence was revised to highlight that phytase had a stronger effect on the P balance (New Lines 573-574).
538 how was modification of the CH fractions measured?
Authors: Regarding the methodology for the starch and hemicellulose analyses, we would like to refer to a sister paper from our group (Vötterl et al., 2019) in which the methods were described.
l.552 pig model for what? It is not really clear what kind of model you try to generate with the pigs. The feed treatments are not relevant for human nutrition or physiology.
Authors: Please see our other comment with respect to the suitability of using the pig as model for humans and the comparability of the diets to human plant-based or vegan diets.
l.554 “excessively” instead of “overly”
Authors: Corrected accordingly (New Lines 607).
l.613ff can the Ca/P ratios in these substrates really indicate homeostasis imbalances?
Authors: We modified this sentence (New Lines 666-668).
l.663 + 667 there were no “dramatic imbalances” in Ca and P homeostasis observed
Authors: “dramatic” was removed (New Line 718).
l.670 vit D, FGF23, ALP as “skeletal disease markers” is not a suitable description, please rephrase
Authors: The phrasing was adapted (New Lines 724-725).
l.671 “eating habits…adolescent pig model” – and again, model for what?
Authors: We modified this sentence. Regarding the animal model we refer to our other comment. Nevertheless, we added a sentence at the end of the Discussion, saying that results may not be transferred one-to-one to humans (New Lines 731-733).
Round 2
Reviewer 2 Report
Dear authors,
thank you for revising the manuscript.
The paper is improved and I think it is a study worth Publishing. However, I do not really understand what model for human nutrition you want to create. If you just leave out this point and describe the pig trial and perhaps also draw species comparisons to rodents, carnivores and omnivores, this would be of high interest.
l. 94ff the reasoning that total P Content was kept constant by phytase Addition is weak - you have Treatment Groups with phytase Addition that are used to compare the phytase effect with the LA effect, which makes sense in the animal Nutrition aspect. You can´t sell this for a human Nutrition model...
l. 132 which nutritional recommendations did you use? Please give a reference (GfE, NRC,...?)
Table 1: delete the double word
l. 361 typo - untreated
l. 607 Are "excessive" amounts of absorbed Ca really excreted via the kidneys in pigs? To my Knowledge, this mechanism is mostly used in hindgut Fermenters
l. 604ff please give the species / species Groups for which These Statements are proven. There are species differences in the dominance of Absorption / excretion routes of Ca and P, as demonstrated by Mack et al. 2015 and Böswald et al. 2018 that should be mentioned in this context.
l. 725 delete blank space
Author Response
Reviewer 2
Dear authors,
thank you for revising the manuscript.
The paper is improved and I think it is a study worth Publishing. However, I do not really understand what model for human nutrition you want to create. If you just leave out this point and describe the pig trial and perhaps also draw species comparisons to rodents, carnivores and omnivores, this would be of high interest.
Authors: Dear reviewer, thank you for this comment. However, there seems to be a misunderstanding. We do not want to create a model for human nutrition as the pig is already widely used as a model for various physiological aspects of humans including digestive physiology. Because the present results have translational aspects to human digestive physiology and bone metabolism, we (as authors) would like to suggest to leave it to the reader to decide if the pig which we selected as the model is suitable as model for human physiology or not.
In any way, we have modified our statement (New Lines 91-97).
- 94ff the reasoning that total P Content was kept constant by phytase Addition is weak - you have Treatment Groups with phytase Addition that are used to compare the phytase effect with the LA effect, which makes sense in the animal Nutrition aspect. You can´t sell this for a human Nutrition model...
Authors: It is true that the addition of microbial phytase is common in animal nutrition. With respect to the interactive effect, this treatment combination was chosen as often different factors in food work synergistically or oppositely. The present diets should be seen as tools to create different P (and Ca) availability along the gastrointestinal tract and in the body after absorption.
- 132 which nutritional recommendations did you use? Please give a reference (GfE, NRC,...?)
Authors: The recommendations refer to the NRC recommendations. The reference number is 47. (New Lines 133-134)
Table 1: delete the double word
Authors: Double word was deleted in Table 1.
- 361 typo – untreated
Author: The spelling was corrected. (New Line 361)
- 607 Are "excessive" amounts of absorbed Ca really excreted via the kidneys in pigs? To my Knowledge, this mechanism is mostly used in hindgut Fermenters
Authors: It was shown in pigs that Ca regulation relies more strongly on the kidney than the intestine. Especially in young pigs, the intestinal uptake of minerals is often higher than the animal would actually need. The excessive minerals (e.g. Ca) are then secreted to the greatest extent in urine. Although pigs are omnivores, they classify as hindgut fermenter as well. The porcine gut has fermentative capacity and the main site of fermentation in pigs is the hindgut.
- 604ff please give the species / species Groups for which These Statements are proven. There are species differences in the dominance of Absorption / excretion routes of Ca and P, as demonstrated by Mack et al. 2015 and Böswald et al. 2018 that should be mentioned in this context.
Author: The species were added as well as the statement that the regulation in other species differs. (New lines 607-614).
- 725 delete blank space
Author: The double space was deleted. (New line 728).